# Mitigating Plasticity Loss through Architectural Design in Continual Learning

Niklas Koeppe [1]   Luiz Felipe Vecchietti [2]   Dongqi Han [3]   Dongsheng Li [3]   Sang Wan Lee [4 5 6]

## Abstract

Neural networks for continual reinforcement learning (CRL) often suffer from plasticity loss, i.e., a progressive decline in their ability to learn new tasks arising from increased representational drift (churn) and Neural Tangent Kernel (NTK) rank collapse. Current methods mitigating this problem involve algorithmic interventions such as regularization, resets, and optimization schedules. Here, we propose InterpLayers, a lightweight architectural solution that combines a fixed, parameter-free reference pathway with a learnable projection pathway using input-dependent interpolation weights. This structure makes InterpLayers orthogonal to existing algorithmic solutions. We show through theoretical analysis that InterpLayers upper-bound the output variability, bound churn, and prevent a collapse of the NTK rank through continual non-zero rank contribution from the interpolation mechanism. Across different distributional shifts, including permutation, windowing, and expansion, InterpLayers outperform similar gated architectures and achieve similar performance as current state-of-the-art methods without the need for optimization-level intervention or the introduction of sensitive hyperparameters. Ablation studies highlight that these improvements are sustained when InterpLayers are combined with existing algorithmic methods for preventing plasticity loss. These results position InterpLayers as a simple, complementary solution for maintaining plasticity in CRL. [1]

## 1. Introduction

Continual reinforcement learning (CRL) requires agents to adapt to a non-stationary stream of tasks without external resets or explicit knowledge of task boundaries. Yet neural networks trained in this setting suffer from *plasticity loss*, i.e., a progressive decline in their ability to learn new tasks overtime despite comparable task complexity and training information, and *catastrophic forgetting*, i.e., a phenomenon where the network loses the ability to perform earlier tasks after learning new ones.

Plasticity loss, our main focus in this work, has been attributed to several interacting factors, including rank collapse of the Neural Tangent Kernel (NTK) (Lyle et al., 2024), unbounded weight growth (Lyle et al., 2023), and representational drift or churn that destabilizes previously acquired knowledge (Tang et al., 2025).

Most existing solutions intervene at the algorithmic level. Reset-based strategies reinitialize parameters on a fixed schedule (Igl et al., 2021; Nikishin et al., 2022; 2023). Continuous plasticity methods modify the optimization process itself, e.g., shrink-perturb (Ash & Adams, 2020), ReDo (Sokar et al., 2023), or regenerative regularization (Kumar et al., 2023). Constraint-based approaches rely on normalization, clipping, or masking to restrict parameter dynamics (Ba et al., 2016; Abbas et al., 2023; Elsayed et al., 2024). While effective, these methods share limitations, including: (i) requiring task boundary information or chosen reset schedules; (ii) introducing hyperparameters such as reset frequencies, perturbation magnitudes, or regularization strengths; (iii) acting externally to the architecture, often outside the optimization framework.

Here, we provide a complementary solution to plasticity loss. Our method works on the architectural level and enhances standard network layers with an additional pathway to build *Interpolation Layers* (InterpLayers). InterpLayers learn a convex interpolator between a skip-like, parameter-free reference pathway and the standard learnable pathway of a neural network. In each layer, the fixed reference pathway maintains a stable representation, while the learnable projection pathway adapts through backpropagation. These two pathways are combined via input-dependent interpolation weights. Due to this dynamic relationship between reference and projection, the network maintains represen-

---

[1] Program of Brain and Cognitive Engineering, KAIST [2] Max Planck Institute for Security and Privacy (MPI-SP) [3] Microsoft Research Asia [4] Center for Neuroscience-inspired AI, KAIST [5] Department of Brain and Cognitive Sciences, KAIST [6] Kim Jaechul Graduate School of AI, KAIST. Correspondence to: Sang Wan Lee <sangwan@kaist.ac.kr>.

*Proceedings of the 43rd International Conference on Machine Learning*, Seoul, South Korea. PMLR 306, 2026. Copyright 2026 by the author(s).

[1] Code available at: https://github.com/n-koeppe/InterpLayer

tational stability without losing its ability to adapt. Unlike ResNet-like skip connections, which only diversify gradient flow, or parameter-efficient tuning methods such as LoRA, which reach computational efficiency through fine-tuning, InterpLayers create a self-regulating mechanism that balances stability and plasticity without external intervention. Moreover, compared to algorithmic approaches like soft-shrink-perturb with layer normalization (Juliani & Ash, 2024), InterpLayers require minimal computational overhead and hyperparameter tuning. Designed as orthogonal components to current solutions for plasticity loss, they can be integrated seamlessly into existing architectures or combined with intervention mechanisms.

InterpLayers are evaluated both theoretically and empirically. We perform a theoretical analysis to investigate how InterpLayers impact churn and NTK rank, showing that InterpLayers intrinsically provide a per-update upper bound to their representational drift and prevent the collapse of NTK rank due to its interpolation mechanisms. For the empirical evaluation, we evaluate the performance of InterpLayers against standard and gated baselines in four ProcGen tasks as described in Juliani & Ash (2024).

We show that InterpLayers achieve similar performance to state-of-the-art algorithmic solutions, such as (soft)-shrink-and-perturb, and that the best performance is obtained when the method is combined with dropout (Srivastava et al., 2014). InterpLayers can also be combined with algorithmic interventions, where they achieve strong performance and improve initial convergence, highlighting that they are an architectural complement to existing solutions. In summary, we show that InterpLayers are effective in preventing plasticity loss and can be a direction for future architectural solutions in continual learning.

Our main contributions can be denoted as follows.

1. We introduce InterpLayers as drop-in replacements for conventional neural network layers. InterpLayers split the layers' input into a reference and a projection pathway that are interpolated to obtain the layer's output.

2. We demonstrate that InterpLayers bound representational drift through controlled interpolation, limit churn growth via pathway stability, and maintain NTK rank under specific assumptions. These guarantees emerge from architectural constraints rather than external interventions.

3. Across four ProcGen environments with different distribution shifts, InterpLayers maintain performance where standard multi-layer perceptron (MLP) layers collapse. Additionally, we show the complementary nature of InterpLayers by their combination with other algorithmic methods that prevent plasticity loss.

## 2. Related works

### 2.1. Algorithmic Approaches to Mitigate Plasticity Loss

**Reset-based interventions.** Periodic parameter reinitialization has often been applied to counter plasticity loss. Igl et al. (2021) proposed resetting only the final layer of a network to preserve learned features while restoring adaptability. Nikishin et al. (2022) showed that resetting selected network parameters on a fixed schedule restores the network's capacity to learn. Later, Nikishin et al. (2023) has shown that resetting the entire network leads to maintenance of plasticity at the cost of losing prior knowledge. To implement these methods, reset schedules and selecting which parameters to reinitialize is needed.

**Continuous plasticity upkeep.** Other methods continuously regulate plasticity during training. Sokar et al. (2023) proposed ReDo, which periodically resets inactive neurons. A continual backpropagation method was presented by Dohare et al. (2024), which adds a step to backpropagation where a small fraction of neurons are continuously reinitialized based on a utility metric. Ash & Adams (2020) applied the shrink-and-perturb methodology to the network where at after optimizer step after they scale down the weights and add noise to maintain plasticity. Farias & Jozefiak (2025) analyzed the effective firing rate of neurons in a network and proposed resetting neurons once they drop below a specific threshold. To prevent unbounded weight drift, Kumar et al. (2023) used regenerative regularization applying L2 penalties to weights. Abbas et al. (2023) showed that increasingly sparse activation patterns decrease gradients, causing plasticity loss. To prevent this, they introduced CReLU as a modified activation function to mitigate sparsity. A regularization technique in Lewandowski et al. (2024) keeps the maximum singular values of each layer close to one, preserving gradient diversity. Moreover, they argue that algorithmic solutions are sensitive to hyperparameters, making it challenging to apply them in generalized settings.

**Normalization and constraint-based methods.** Other approaches alleviate plasticity loss by constraining the network dynamics. Lyle et al. (2023) showed that LayerNorm can slow down plasticity loss, as it helps to maintain NTK rank. Elsayed et al. (2024) investigated weight clipping to provide an upper bound to parameter growth. To stabilize optimization, Miyato et al. (2018) have shown that spectral normalization can constrain Lipschitz constants. Although plasticity loss is reduced, representational capacity is affected by the constraints added by these methods. Lee et al. (2025) proposed an architecture called SimbaV2, which constrains weight growth and feature norm by hyperspherical normalization and makes use of reward scaling to maintain gradient stability. Nauman et al. (2024) introduced a method which combines LayerNorm, weight decay, and full-parameter resets to scale a SAC policy to seven times

its size, improving performance while maintaining sample efficiency. Lately, a new mechanism that dynamically regulates the weights of samples in the replay buffer to sustain gradient diversity has also been proposed to prevent NTK rank collapse and sustain plasticity in Wu et al. (2026).

## 2.2. Architectural Mechanisms for Stability in Neural Networks

Various innovations in neural network architectures have been proposed to balance stability and plasticity, even though they have not been directly applied to continual learning. Skip connections and residual pathways have been vastly investigated to improve gradient flow and regulate information propagation in computer vision models (He et al., 2016; Srivastava et al., 2015). Gating mechanisms that control information flow have likewise been shown to be highly effective in natural language processing architectures (Hochreiter & Schmidhuber, 1997; Cho et al., 2014). Lately, Qiu et al. (2025) has introduced input-dependent gating within attention mechanisms, allowing the network to suppress unnecessary information dynamically. A data-dependent variable depth approach to decrease computational load has been proposed by Bae et al. (2025), introducing the ability to decrease model size while improving few-shot accuracy. Networks that generate specific parameters conditioned on input features, such as HyperNetworks Ha et al. (2016), have also been investigated to introduce architectural flexibility in meta-learning tasks.

## 2.3. Theoretical Understanding of Plasticity Loss

Another set of works explored key theoretical features to improve the understanding of plasticity loss in neural networks. Lyle et al. (2024) showed that the effective NTK rank is strongly related to the ability of the network to adapt in a continual learning setting. Specifically, they demonstrate that NTK rank collapse correlates with a decrease in performance. The unconstrained drift of network parameters away from their initialization has also been described as a cause for plasticity loss in CRL by Kumar et al. (2023). The instability of network outputs, i.e., *churn*, is investigated by Tang & Berseth (2024); Tang et al. (2025) as an important factor in plasticity loss. In addition to these metrics, Lewandowski et al. (2023) showed that a decrease in curvature directions is another indicator of plasticity loss in neural networks. Based on these findings, we theoretically investigate the effects of InterpLayers on churn and effective NTK rank.

## 3. Methods

### 3.1. Preliminaries

We consider an agent that learns in a CRL environment interacting with a sequence of tasks $\{\mathcal{M}_1, \mathcal{M}_2, ..., \mathcal{M}_K\}$

following a Markov Decision Process (MDP), where each $\mathcal{M}_i = (\mathcal{S}_i, \mathcal{A}_i, P_i, r_i, \gamma)$ may have different state spaces $\mathcal{S}_i$, action spaces $\mathcal{A}_i$, transition dynamics $P_i$, and reward functions $r_i$. The variable $\gamma$ is a discount factor. The tasks are separated by distribution shifts, which can range from small changes, e.g., reinitializing the environment with a new random seed, to substantial changes, e.g., permutations on the observation axis that completely modify the input distribution. At each timestep $t$, the agent observes state $s_t$, selects action $a_t$ according to policy $\pi_\theta(a|s)$, receives reward $r_t$, and transitions to state $s_{t+1}$. The policy $\pi_\theta(a|s)$ is parameterized by a neural network with weight parameters $\theta$ and trained via backpropagation.

In our CRL setting, the current task $\mathcal{M}$ is changed after a fixed number of environment steps. The agent is given no information about the task boundaries or identities of the task, so it does not know which task it has to solve at a given time. The agent should adapt to a new task by modifying its set of parameters $\theta$ online, having a shared policy for multiple tasks. The policy does not store past experiences in another data structure to sample during training. In this way, the policy should maintain a balance between stability (preserving knowledge) and plasticity (acquiring new knowledge) in a non-stationary environment.

### 3.2. The Interpolation Layer

As an architectural solution to tackle plasticity loss, we introduce InterpLayers (Figure 1), which are task-agnostic, do not require additional hyperparameters, can be seamlessly integrated into existing neural network architectures, and can be combined with existing algorithmic approaches that mitigate plasticity loss.

**Core mechanism.** Each InterpLayer obtains intermediate outputs for two complementary pathways: (i) a *reference pathway* given by a fixed, parameter-free mapping (identity, sparse selection, or padding if dimensions differ); and (ii) a *projection pathway* with standard learnable parameters. Learnable interpolation weights then combine both outputs, allowing the network to learn when to rely on preservation and when to adapt. Mathematically, given an input $\mathbf{x} \in \mathbb{R}^d$, the InterpLayer output is given as

$$\mathbf{h}(\mathbf{x}) = (1 - z(\mathbf{x})) \odot h_{\text{ref}}(\mathbf{x}) + z(\mathbf{x}) \odot h_{\text{proj}}(\mathbf{x}), \quad (1)$$

where $\odot$ denotes element-wise multiplication and $h_{\text{ref}}, h_{\text{proj}}$, and $z(\mathbf{x})$ are defined as

$$h_{\text{ref}}(\mathbf{x}) = \mathbf{P}\mathbf{x}, \quad (\mathbf{P} = \mathbf{I} \text{ when } d_{\text{in}} = d_{\text{out}}), \quad (2)$$

$$h_{\text{proj}}(\mathbf{x}) = \phi(\mathbf{W}_p\mathbf{x} + \mathbf{b}_p), \quad \mathbf{W}_p, \mathbf{b}_p \text{ (learnable)}, \quad (3)$$

$$z(\mathbf{x}) = \sigma(\mathbf{W}_i\mathbf{x} + \mathbf{b}_i), \quad \mathbf{W}_i, \mathbf{b}_i \text{ (learnable)}, \quad (4)$$

$d_{\text{in}}$ and $d_{\text{out}}$ denote the input and output dimensionalities of the layer, $\phi$ is a non-linear activation function, and $\sigma$ is a sigmoid layer.

*Figure 1.* **The InterpLayer Architecture.** The input $\mathbf{x}$ is processed through a fixed reference pathway $h_{\text{ref}}(\mathbf{x})$ and a learnable projection pathway $h_{\text{proj}}(\mathbf{x})$. The learnable interpolation weights $z(\mathbf{x})$ dynamically interpolate the outputs from both pathways to produce the output $\mathbf{h}(\mathbf{x})$.

**Definition of the individual structures.** The reference pathway functions as a parameter-free deterministic module $P$ that preserves the geometric structure of the input. For linear layers, we implement $P$ using an *IdentityProject* block: if $d_{\text{in}} = d_{\text{out}}$, $P$ is the identity; if $d_{\text{out}} < d_{\text{in}}$, $P$ is a fixed Johnson–Lindenstrauss (Dasgupta & Gupta, 2003) projection with orthonormal rows ($PP^\top = I_{d_{\text{out}}}$) constructed with a seed fixed per layer; and if $d_{\text{out}} > d_{\text{in}}$, $P$ performs zero-padding to preserve the identity structure. For convolutional layers, we use an *IdentityDownsample* block: when the spatial resolution changes (i.e., stride $> 1$), we use average pooling; when channel counts differ, we apply channel slicing (if $c_{out} < c_{in}$) or padding (if $c_{out} > c_{in}$). These modules do not have learnable parameters, remain fixed during training, and serve only to preserve spatial structure to have a stable reference for interpolation.

In contrast, the projection pathway enables adaptation through standard learning, similar to an MLP layer. The interpolation weights $z(\mathbf{x}) \in (0, 1)^h$ regulate the contribution of reference and projection, providing the network with a dynamic preservation–adaptation tradeoff. This mechanism is similar to input gates in GRUs (Cho et al., 2014), but has a key difference: $h_{\text{ref}}$ is a fixed skip from the current input rather than a recurrent hidden state from the past. Intuitively, if the mean of the interpolation weights is closer to 0, the output $\mathbf{h}(\mathbf{x})$ is mostly represented by the reference; oppositely, if the mean is closer to 1, $\mathbf{h}(\mathbf{x})$ is mostly represented by the output of the projection.

**Integration to convolutional layers.** InterpLayers can replace standard MLP layers following Eqs. (1)-(4). For convolutional layers processing image data $\mathbf{X} \in \mathbb{R}^{C_{\text{in}} \times H \times W}$ as part of the state space, $h_{\text{ref}}$, $h_{\text{proj}}$, and $z(\mathbf{x})$ are defined as

$$h_{\text{ref}}(\mathbf{X}) = \mathbf{P}_r * \mathbf{X} \tag{5}$$
$$h_{\text{proj}}(\mathbf{X}) = \phi(\mathbf{W}_p * \mathbf{X} + \mathbf{b}_p), \tag{6}$$
$$z(\mathbf{X}) = \sigma(\mathbf{W}_i \cdot \beta(\mathbf{X}) + \mathbf{b}_i), \tag{7}$$

where $*$ denotes a convolution operation and $\beta$ is a global average pooling operation to produce channel-wise interpolation.

### 3.3. Theoretical Properties of InterpLayers

We analyze the mathematical properties of InterpLayers, focusing on two key properties: bounded representational drift and preservation of gradient diversity.

#### 3.3.1. BOUNDED REPRESENTATIONAL DRIFT

The dual-pathway structure of InterpLayers ensures that changes in the output remain bounded under parameter updates. For an update $\Delta\theta = (\Delta\theta_p, \Delta\theta_z)$, the first-order output change is given as

$$\Delta h(\mathbf{x}) = z(\mathbf{x}) \odot \Delta h_{\text{proj}}(\mathbf{x}) + \Delta z(\mathbf{x}) \odot [h_{\text{proj}}(\mathbf{x}) - h_{\text{ref}}(\mathbf{x})]. \tag{8}$$

This decomposition shows that updates are constrained. The update of the projection pathway is modulated by the interpolation weights $z(\mathbf{x}) \in (0, 1)^h$, while the interpolation update is bounded by the difference between the reference and projection.

**Theorem 3.1** (Bounded Output Variability). *If $h_{proj}$ is $L_p$-Lipschitz in its parameters $\theta_p$ and $z$ is $L_z$-Lipschitz in $\theta_z$, then*

$$\|\Delta h(\mathbf{x})\|_2 \leq \|z(\mathbf{x})\|_\infty L_p \|\Delta\theta_p\|_2 + L_z \|\Delta\theta_z\|_2 D(\mathbf{x}), \tag{9}$$

*where* $D(\mathbf{x}) = \|h_{proj}(\mathbf{x}) - h_{ref}(\mathbf{x})\|_2$.

The proof is deferred to Appendix A.1. This bound implies that, assuming finite-horizon update magnitudes, churn is upper-bounded by a polynomial function of the training horizon. Compared to standard MLP layers where churn can grow with increasing parameter norms. This theorem makes use of the fact that the reference pathway is parameter-free

at the layer level, such that only projection and interpolation weights contribute to the representational drift.

### 3.3.2. GRADIENT DIVERSITY PRESERVATION

InterpLayers preserve gradient diversity by altering the structure of the NTK. Given the InterpLayer formulation, the gradient with respect to network parameters decomposes as

$$\nabla_\theta h(x) = \begin{bmatrix} z(x) \odot \nabla_{\theta_p} h_{\mathrm{proj}}(x) \\ \nabla_{\theta_z} z(x) \odot \left( h_{\mathrm{proj}}(x) - h_{\mathrm{ref}}(x) \right) \end{bmatrix}. \quad (10)$$

This yields an NTK with separate projection and interpolation contributions, where $N_{\mathrm{proj}}$ and $N_{\mathrm{interp}}$ denote the contributions to the NTK from projection and interpolation parameters, respectively. Intuitively, the interpolation mechanism adds a gradient component through the interpolation parameters, which can sustain diversity in the NTK when the corresponding interpolation-gradient features are nonzero and diverse across samples, even if the projection pathway degenerates. For readers unfamiliar with NTK calculations, we provide a step-by-step derivation and empirical estimator details in Appendix A.3.

**Theorem 3.2** (Rank contribution of the interpolation pathway). *Consider a finite sample set $\{x_i\}_{i=1}^n$. Let*

$$J_p(x_i) := \frac{\partial h_{\mathrm{proj}}(x_i)}{\partial \theta_p}, \qquad J_z(x_i) := \frac{\partial z(x_i)}{\partial \theta_z},$$

*and define the sample-wise projection and interpolation Jacobian features*

$$u_i := \mathrm{vec}\left( z(x_i) \odot J_p(x_i) \right),$$
$$v_i := \mathrm{vec}\left( J_z(x_i) \odot \left( h_{\mathrm{proj}}(x_i) - h_{\mathrm{ref}}(x_i) \right) \right).$$

*Let $U$ and $V$ be the matrices whose $i$-th rows are $u_i^\top$ and $v_i^\top$, respectively. Then the empirical NTK of an InterpLayer decomposes as*

$$N_{\mathrm{IL}} = UU^\top + VV^\top, \qquad N_{\mathrm{interp}} = VV^\top.$$

*Consequently,*

$$\mathrm{rank}(N_{\mathrm{IL}}) \geq \mathrm{rank}(N_{\mathrm{interp}}) = \mathrm{rank}(V).$$

*In particular, if*

$$\sum_{i=1}^n \|v_i\|_2^2 > 0,$$

*then*

$$\mathrm{rank}(N_{\mathrm{interp}}) \geq 1.$$

*More generally, if $r$ of the vectors $\{v_i\}_{i=1}^n$ are linearly independent, then*

$$\mathrm{rank}(N_{\mathrm{interp}}) \geq r.$$

Theorem 3.2 shows that the interpolation pathway contributes an additional positive semidefinite Gram component to the empirical NTK. Intuitively, as long as the interpolation-gradient features $v_i$ are nonzero and linearly diverse across samples, the interpolation pathway contributes corresponding rank to the NTK. Empirical verification of NTK rank during training is provided in Appendix L.3.

## 4. Results

### 4.1. Experimental Setup

We employ the ProcGen environment (Cobbe et al., 2020) to evaluate the proposed framework in CRL settings. As benchmark tasks, we apply three distribution shifts previously introduced by Juliani & Ash (2024) on the *Coinrun*, *Jumper*, *Fruitbot*, and *Heist* environments (example visualizations of the shifts are shown in Appendix I). These three variations are named *permute*, *window*, and *expand*. For the *permute* task, at each shift point, we randomly permute all pixels in the observation space. In the *window* task, we change the random seed that is used to generate the levels at each shift point. In the *expand* task, we start training with 100 levels, and at each shift point, we extend the training set by increments of 100, ending with 1000 levels after the final shift.

**InterpLayer Baseline.** The network used in our experiments consists of an encoder with four convolutional layers followed by a linear layer. To select an appropriate baseline, we evaluated multiple architectural variants of InterpLayers, including versions that replace all encoder layers with InterpLayers as well as variants that combine lighter interventions such as dropout (Srivastava et al., 2014). Results for these variants are shown in Appendix M. Based on this analysis, we select the version that replaces only the convolutional layers in the encoder with InterpLayers and applies dropout on the projection pathway during training. We call this variant **InterpLayers** across the experiments. Empirically, we observe that using dropout in the projection increases the activation-level variance in the projection pathway, thereby enlarging the representational gap between reference and projection. We hypothesize that this effect increases the benefits of our proposed interpolation mechanisms.

Training is performed using Proximal Policy Optimization (PPO) (Schulman et al., 2017). When compared to standard networks, InterpLayers introduce additional learnable parameters in the interpolation mechanism. For a fair comparison, we compare InterpLayers with architectures using a similar number of parameters. Additionally, the baselines are tested with and without dropout, keeping the variant that achieves superior plasticity retention. Details regarding the training details and computational cost comparison are

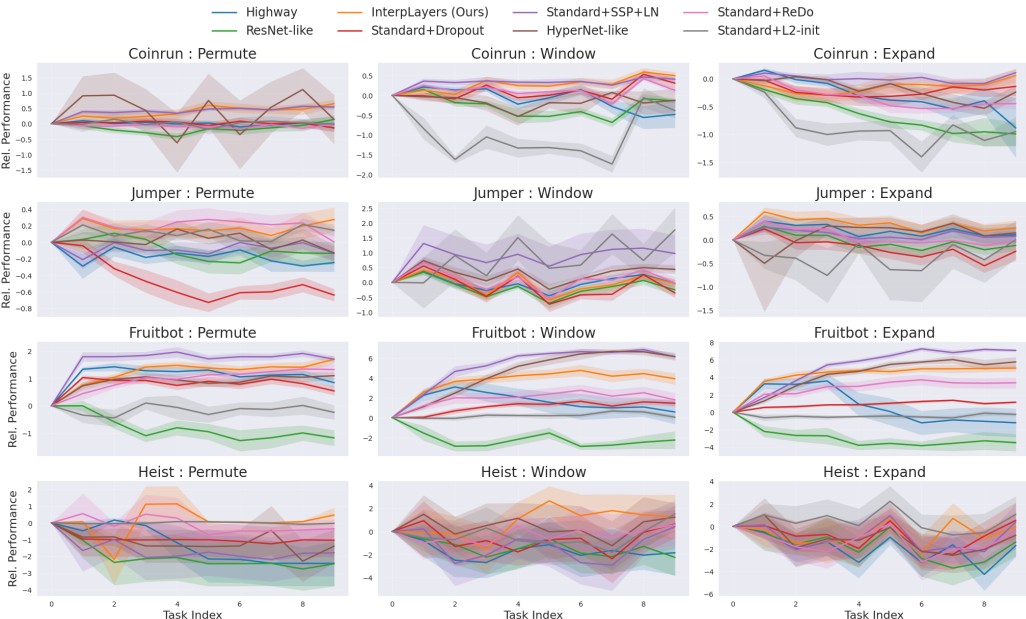

*Figure 2.* **Performance (relative to the initial task) for InterpLayers and baselines under three distribution shifts for four ProcGen environments.** Performance is defined as the mean reward over the final 50 episodes of each task, normalized relative to the initial task, with shaded regions denoting standard error across 10 seeds.

given in Appendix B and C, respectively. Our method is compared with three architectural baselines: a ResNet-like architecture (He et al., 2016), Highway Networks (Srivastava et al., 2015), and a HyperNet-like architecture (Ha et al., 2016); and against strong algorithmic baselines proposed and benchmarked in Juliani & Ash (2024): soft shrink-perturb with layer norm (SSP+LN), ReDo, and L2-init. SSP+LN mixes the current weight with initialization noise after each optimizer step (check Appendix D for implementation details). Finally, we investigate the performance of InterpLayers combined with algorithmic methods, showcasing their potential to complement other approaches that sustain plasticity. The results are average runs of 10 random seeds where training is performed for 50,000 epochs, with distribution shifts being applied every 5,000 epochs.

### 4.2. InterpLayer Performance Under Distribution Shifts

We evaluate whether InterpLayers can maintain performance across sequential tasks separated by distribution shifts. Fig. 2 shows the normalized performance, defined as the mean reward over the final 50 episodes of each task, normalized relative to the initial task and plotted with shaded regions denoting the standard error across 10 seeds for eight network variants: InterpLayers, Highway, ResNet-like, HyperNet-like, as well as the baselines, Standard with Dropout, Standard with SSP+LN, Standard with ReDo, and Standard with L2-init. For all dropout variants, we set the dropout rate to 0.05.

**Permute**: Compared to other shifts, this shift requires full representational relearning. The ResNet-like network loses performance after the initial tasks, dropping below 0 relative to the initial task in all environments. The performance of Highway Networks also decreases for Coinrun, Heist, and Jumper. HyperNet-like networks show very high variance and do not maintain their performance consistently. Standard+ReDo shows consistent performance retention but remains overall slightly worse than InterpLayers and SSP+LN. Standard+L2-init shows strong retention in all environments except Fruitbot. Our proposed InterpLayer achieves the best final performance. InterpLayers and SSP+LN remain above zero and are the top-performing methods for most tasks.

**Window**: Changing to newly generated levels at each shift produces a clear performance separation. InterpLayers and SSP+LN consistently achieve better performance, whereas ResNet-like and Highway networks exhibit plasticity loss across the four environments. The Standard+Dropout baseline achieves good performance in Coinrun and Heist, but does not outperform InterpLayers and SSP+LN. Standard+L2-init shows good performance in some environments but weak performance in Coinrun, while the ReDo and HyperNet-like baselines have stable performances but are often outperformed by SSP+LN and InterpLayers.

**Expand**: Increasing the number of levels provides a gradual adaptation challenge. Consistent with the results for permute and window, InterpLayers and SSP+LN achieve the

| | Performance Retention at Final Task | | | | | | | | | | | |
| | CoinRun | | | Jumper | | | FruitBot | | | Heist | | |
| Method | Perm | Win | Exp | Perm | Win | Exp | Perm | Win | Exp | Perm | Win | Exp |
| Standard+SSP+LN | 0.546 | 0.406 | **0.111** | -0.133 | **0.973** | 0.077 | **1.706** | **6.129** | **7.069** | -1.787 | 0.407 | **0.375** |
| InterpLayers (Ours) | **0.655** | **0.498** | 0.059 | **0.276** | -0.045 | **0.254** | 1.703 | 3.916 | 5.050 | **0.465** | **1.253** | -0.056 |

*Table 1.* Performance retention at the final task for Standard+SSP+LN and InterpLayers. Values report mean relative performance at task index 9, normalized relative to the initial task. Higher values indicate better retention. Best performance per column is shown in bold.

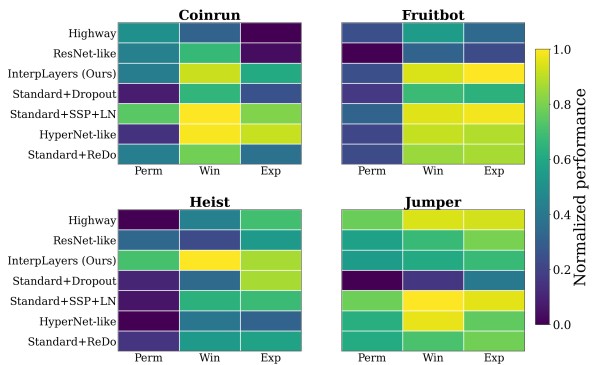

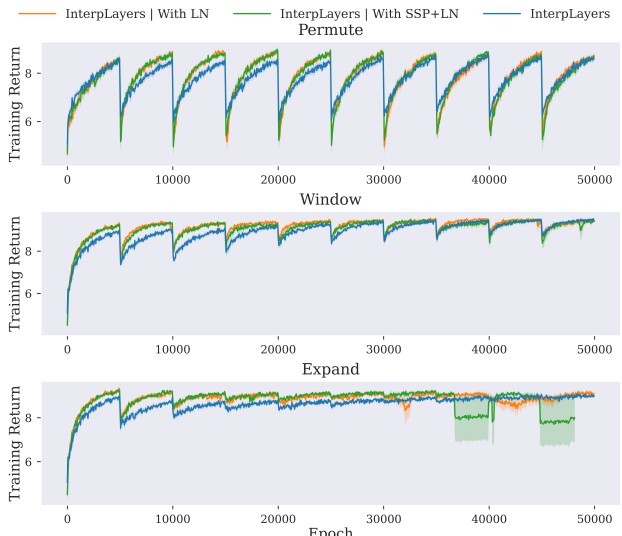

*Figure 3.* **Raw Rewards (normalized for each task) for InterpLayers and baselines under three distribution shifts for four ProcGen environments at the end of the training.** Across all shift types, InterpLayers and SSP+LN achieve the highest raw rewards.

*Figure 4.* **Ablation study combining InterpLayers with Layer-Norm and SSP-LN for Coinrun.** Combining InterpLayers with these methods improves its convergence speed for initial tasks.

best performance for Coinrun, Fruitbot, and Jumper. In this shift, Standard+L2-init shows weak performance in almost all environments. ReDo and HyperNet-like show consistent performance but remain overall worse than InterpLayers and SSP+LN. For Jumper, most methods achieve similar performance curves with final values dropping below 0. This suggests that generalization is harder in this task.

Across all shift types, InterpLayer networks outperform standard architectural baselines, including Standard+Dropout, Highway, ResNet-like, and HyperNet-like networks. Table 1 compares the two best-performing methods: SSP+LN and InterpLayers. They achieve comparable performance across all environments and shift types. Unlike SSP+LN, however, InterpLayers do not apply optimization-level interventions and sustain plasticity through architectural design. Compared to other algorithmic baselines such as ReDo and L2-init, InterpLayers show stronger overall retention. This supports the application of InterpLayers in general settings in which it is challenging to fulfill the constraints introduced by algorithmic methods.

We also show the normalized raw rewards obtained by different methods during training in Fig. 3. It is seen that SSP+LN and InterpLayers achieve similar performance regarding raw rewards at the end of training, with InterpLayers showcasing more stable performance than other baselines.

### 4.3. InterpLayers with Algorithmic Interventions

To evaluate InterpLayers as an orthogonal solution to algorithm approaches, we combine InterpLayers with LayerNorm (LN) and SSP+LN. Figure 4 shows the raw performance of the combined methods. The results show that combining InterpLayers with LN improves the initial convergence speed of InterpLayers, increasing its performance for the first tasks. Notably, InterpLayers with SSP+LN yield the same performance as InterpLayers combined with LN.

### 4.4. Empirical Validation of Theoretical Properties

We show the empirical validation for churn in Fig. 5 for the Coinrun environment. Further details on the methodology for calculating this metric are provided in Appendix H. We observe that InterpLayers maintain a low representational drift and reduce churn during training. Adding dropout to the standard baseline also reduces churn, suggesting that dropout-induced mechanisms are effective in slowing plasticity loss. Other architectural baselines, ResNet-like, HyperNet-like, and Highway Networks, have the highest

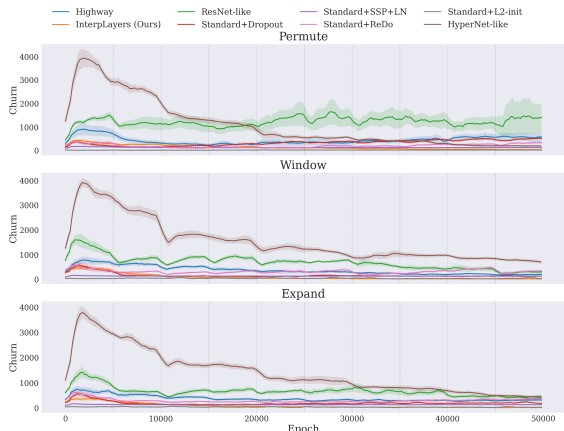

*Figure 5.* **Evolution of churn for InterpLayers and baselines under distribution shifts in Coinrun.** Shaded regions denote variability across 10 seeds, and vertical lines indicate shift points. Across all conditions, InterpLayers maintain lower churn compared to the standard baselines.

churn, with HyperNet-like networks exhibiting the largest early drift. SSP+LN maintains a stable churn throughout training, while ReDo and L2-init also remain comparatively low. These observations are consistent across all distribution shifts.

### 4.5. Analyzing the Interpolation Mechanism

Figure 6 shows the per-layer distributions of the interpolation weights averaged for early training (tasks 1-5) and late training (tasks 6-10). It is observed that early training is characterized by high variance and broad weight distributions. In late training, the distributions shift towards the reference pathway, which indicates that low-level features are stabilized. We see that the average value for interpolation weights is around 0.2. This pattern is more prevalent in the expand and window tasks, while less prevalent in the permute task.

## 5. Discussion

The analysis presented in Fig. 6 shows that InterpLayers implicitly develop a hierarchical structure. While fixed interpolation weights of $z = 0.5$ would act like ResNet-like skip connections, we observe a different pattern. This distinction is central: residual layers use a fixed additive skip connection, whereas InterpLayers learn an input-dependent convex interpolator between the fixed reference pathway and the learnable projection pathway. Across all task shifts, the interpolation weights do not saturate towards 0, 0.5, or 1; instead, their mean across all layers settles around $z \approx 0.2$, which indicates a preference for the reference pathway. Together with the results of the ResNet-like baseline in Fig. 2, this suggests that the benefits of InterpLayers are not due

to the additional skip pathway alone, but also stem from the learnable interpolation mechanism. Similar findings have been shown on gated attention mechanisms, where gates suppress up to 88% of the contextual information (Qiu et al., 2025). In both cases, learned gating does not degenerate into a binary decision, which suggests that neural networks favor a meaningful structured representation over reparameterization if given the choice.

In our case, this behavior evolves naturally due to the input-specific interpolation mechanism. We also observe this kind of self-organization in other fields of deep learning, where lower layers act as feature extractors while higher layers adapt to more task-specific details (Yosinski et al., 2014).

The evolution of metrics related to theoretical properties (Section 3.3) is crucial to mitigate plasticity loss. Our empirical results for churn evolution (Fig. 5) show that it decreases over time using InterpLayers. These results agree with results recently presented by Tang et al. (2025), demonstrating that reducing churn is important to keep plasticity in neural networks.

Standardized proxy metrics for plasticity loss remain an open problem. Earlier works have proposed alternatives such as probe-based plasticity (Lyle et al., 2023) and sample plasticity (Elsayed et al., 2024), while recent work highlights churn and NTK rank as useful diagnostics for plasticity loss (Lyle et al., 2024; Tang et al., 2025). Here, we focus our analysis on performance retention with churn and encoder-side gradient-rank proxies. These findings empirically verify the theoretical advantages of using InterpLayers in CRL.

Furthermore, the analysis in Section 3.3 suggests that the plasticity of the network can be estimated through the joint behavior of the interpolation weights $z$ and the representational gap $D$ defined in Theorem 3.1. Together, $z$ and $D$ indicate how much a layer can adapt, with $z$ controlling the exposure to the projection output and $D$ controlling sensitivity to the projection-reference discrepancy. This analysis helps to understand why applying dropout to the projection enhances InterpLayers performance. Dropout stochastically masks the projection activations, which increases activation-level variance and prevents the projection from aligning with the reference, sustaining a non-zero $D$. Following this, gradients of the interpolation weights remain active for the entire learning period, mitigating plasticity loss. Thus, we view dropout as an active tool that interacts with the projection and interpolation dynamics of InterpLayers. We further discuss the relationship between InterpLayers and dropout in Appendix J.

Architectures with gated mechanisms (Hochreiter & Schmidhuber, 1997; Cho et al., 2014) and residual networks (He et al., 2016; Srivastava et al., 2015) have been responsible for key advances in recurrent neural networks, con-

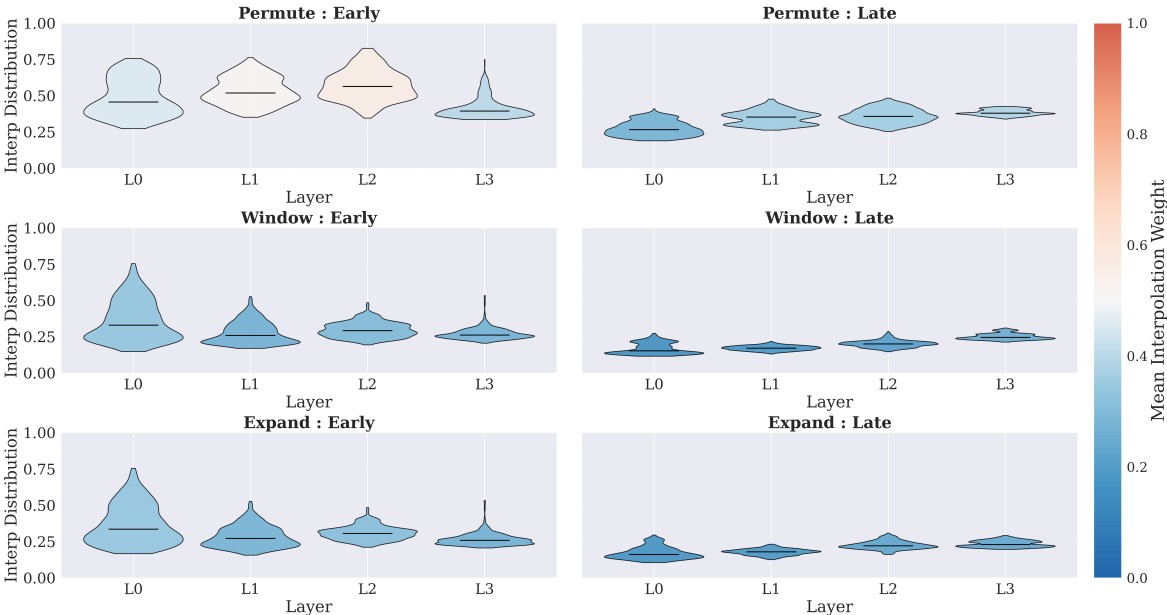

*Figure 6.* **Distribution per-layer of interpolation weights in early training (tasks 1-5) and late training (tasks 6-10).** Interpolation weights are initialized with a mean equal to 0.5. We can see that at the end of the training, these weights saturate with values around 0.2, with permute achieving higher average values when compared to window and expand.

volutional neural networks, and transformer models (Qiu et al., 2025). In the same direction, InterpLayers present an interpolation mechanism that sustains plasticity through different streams and gated interventions. This can be viewed as a form of adaptive-computation, and at a high-level, can be related to variable-depth methods that achieve adaptation through depth. In addition, they provide a complementary architectural axis to other plasticity methods. This resonates with models inspired by neuroscience in which dendritic compartments and gating mechanisms solve the stability-plasticity trade-off in cortical circuits (Bengio et al., 2015; Urbanczik & Senn, 2014). Our findings place InterpLayers as a simple but general mechanism that enriches the toolbox of CRL toward architectures that implicitly solve plasticity loss.

## 6. Conclusion

In this paper, we introduce InterpLayers as an architectural solution to plasticity loss in CRL. Requiring no algorithmic changes or sensitive hyperparameters, InterpLayers provide continuous regulation of plasticity through a dual-pathway design. Our findings show that InterpLayers mitigate plasticity loss in four ProcGen environments. We show that combining InterpLayers with dropout improves its performance, obtaining results comparable to state-of-the-art methods. These findings suggest that characteristics learned by dropout regularization help the interpolation dynamics of InterpLayers. InterpLayers can also be combined with existing algorithmic techniques to maximize performance.

Important future research directions include investigating the performance of InterpLayers when having different levels of sparsity in the network and how this architectural mechanism can be extended to larger networks to sustain plasticity in general scenarios.

## Impact Statement

Our work impacts the development of models that learn continuously. Even though the application of these methodologies has been limited to large policies such as large language models, these may start posing risks in the future. These risks include using these systems to manipulate and persuade users by learning through interaction. We reinforce that our main goal with our work is to advance the understanding of neural networks in CRL and propose neuroscience-inspired AI architectures.

## Acknowledgments

This research was supported by the MSIT (Ministry of Science, ICT), Korea, under the Global Research Support Program in the Digital Field program (RS-2024-00436680) supervised by the IITP (Institute for Information & Communications Technology Planning & Evaluation). This project is supported by Microsoft Research Asia. This work was supported by the Institute of Information & communications Technology Planning & Evaluation (IITP) grant funded by

the Korean government (MSIT) (No. RS-2023-00233251, System3 reinforcement learning with high-level brain functions). This work was supported by the National Research Foundation of Korea (NRF) grant funded by the Korean government (MSIT) (No. RS-2024-00341805). This work was supported by Institute for Information & communications Technology Planning & Evaluation (IITP) grant funded by the Korean government (MSIT) (RS-2019-II190075, Artificial Intelligence Graduate School Program (KAIST)).

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

# A. Theoretical Properties: Proofs and Extensions

## A.1. Proof of Theorem 3.1

Starting from the first-order output change (Eq. 8):

$$\Delta h(\mathbf{x}) = z(\mathbf{x}) \odot \Delta h_{\text{proj}}(\mathbf{x}) + \Delta z(\mathbf{x}) \odot [h_{\text{proj}}(\mathbf{x}) - h_{\text{ref}}(\mathbf{x})]. \tag{11}$$

By the triangle inequality and the property $\|a \odot b\|_2 \leq \|a\|_\infty \|b\|_2$:

$$\|\Delta h(\mathbf{x})\|_2 \leq \|z(\mathbf{x})\|_\infty \|\Delta h_{\text{proj}}(\mathbf{x})\|_2 + \|\Delta z(\mathbf{x})\|_\infty \cdot D(\mathbf{x}), \tag{12}$$

where $D(\mathbf{x}) = \|h_{\text{proj}}(\mathbf{x}) - h_{\text{ref}}(\mathbf{x})\|_2$.

By Lipschitz continuity assumptions:

$$\|\Delta h_{\text{proj}}(\mathbf{x})\|_2 \leq L_p \|\Delta\theta_p\|_2, \tag{13}$$
$$\|\Delta z(\mathbf{x})\|_\infty \leq L_z \|\Delta\theta_z\|_2. \tag{14}$$

Therefore:

$$\|\Delta h(\mathbf{x})\|_2 \leq \|z(\mathbf{x})\|_\infty L_p \|\Delta\theta_p\|_2 + L_z \|\Delta\theta_z\|_2 D(\mathbf{x}). \tag{15}$$

Since $z(\mathbf{x}) \in (0,1)^h$ due to the sigmoid, $\|z(\mathbf{x})\|_\infty < 1$. This completes the proof of the per-update first-order output sensitivity bound. $\square$

## A.2. Corollary: Finite-Time Churn Growth Bound

Consider a sequence of updates $\{\theta_t\}_{t=0}^T$ under learning rate $\eta$. By Theorem 3.1, each step incurs an output change bounded by

$$\|\Delta h_t(\mathbf{x})\|_2 \leq \eta \left( \|z_t(\mathbf{x})\|_\infty L_p \|\nabla_{\theta_p}\mathcal{L}_t\|_2 + L_z \|\nabla_{\theta_z}\mathcal{L}_t\|_2 D_t(\mathbf{x}) \right) \tag{16}$$

where $D_t(\mathbf{x}) = \|h_{\text{proj},t}(\mathbf{x}) - h_{\text{ref}}(\mathbf{x})\|_2$. Accumulating over $t$ and applying the triangle inequality yields

$$\|h_{\theta_T}(\mathbf{x}) - h_{\theta_0}(\mathbf{x})\|_2 \leq BT, \tag{17}$$

for a constant $B$ depending on $\eta$, $L_p$, $L_z$, the finite-horizon gradient magnitudes, and the finite-horizon values of $D_t(\mathbf{x})$. Squaring and taking expectation over $\mathcal{D}_{\text{ref}}$ gives

$$\mathcal{C}_T \leq B^2 T^2, \tag{18}$$

establishing a finite-time quadratic upper bound on churn. This bound should be interpreted as a growth bound over a fixed training horizon $T$, rather than a time-uniform stability guarantee.

## A.3. Proof of Theorem 3.2

Starting from Eq. (10), the Jacobian of the InterpLayer output with respect to its parameters decomposes for each sample $x_i$ as

$$\frac{\partial h(x_i)}{\partial \theta} = \begin{bmatrix} z(x_i) \odot J_p(x_i) \\ J_z(x_i) \odot \left( h_{\text{proj}}(x_i) - h_{\text{ref}}(x_i) \right) \end{bmatrix}.$$

Vectorizing the two blocks gives the Jacobian feature vectors $u_i$ and $v_i$ defined in the theorem.

Therefore, the empirical NTK entry between samples $x_i$ and $x_j$ is

$$N_{\text{IL}}(x_i, x_j) = \langle u_i, u_j \rangle + \langle v_i, v_j \rangle.$$

If $U$ and $V$ collect the row vectors $u_i^\top$ and $v_i^\top$, respectively, then the empirical InterpLayer NTK and its interpolation component are defined as

$$N_{\text{IL}} := UU^\top + VV^\top, \qquad N_{\text{interp}} := VV^\top.$$

Since $VV^\top$ is a Gram matrix, it is positive semidefinite and

$$\text{rank}(N_{\text{interp}}) = \text{rank}(VV^\top) = \text{rank}(V).$$

For any $\alpha \in \mathbb{R}^n$,

$$\alpha^\top(UU^\top + VV^\top)\alpha = \|U^\top\alpha\|_2^2 + \|V^\top\alpha\|_2^2.$$

Therefore,

$$\alpha \in \ker(UU^\top + VV^\top) \iff U^\top\alpha = 0 \text{ and } V^\top\alpha = 0,$$

so

$$\ker(UU^\top + VV^\top) = \ker(UU^\top) \cap \ker(VV^\top).$$

Hence

$$\ker(N_{\text{IL}}) \subseteq \ker(N_{\text{interp}}),$$

so

$$\text{nullity}(N_{\text{IL}}) \leq \text{nullity}(N_{\text{interp}}).$$

By rank–nullity, this implies

$$\text{rank}(N_{\text{IL}}) \geq \text{rank}(N_{\text{interp}}).$$

If $\sum_{i=1}^n \|v_i\|_2^2 > 0$, then

$$\text{tr}(N_{\text{interp}}) = \text{tr}(VV^\top) = \sum_{i=1}^n \|v_i\|_2^2 > 0,$$

so $N_{\text{interp}} \neq 0$, which implies

$$\text{rank}(N_{\text{interp}}) \geq 1.$$

Finally, if $r$ of the vectors $\{v_i\}_{i=1}^n$ are linearly independent, then $\text{rank}(V) \geq r$, and therefore

$$\text{rank}(N_{\text{interp}}) \geq r.$$

This completes the proof. $\qquad\square$

### A.4. Weight Norm Regularization

Although not central to the main text, we note that interpolation gates implicitly regularize effective weight norms. Define the effective contribution at time $t$ as

$$\|\mathbf{W}_{\text{eff}}(t)\|_F^2 \leq \mathbb{E}_x[\|z_t(x)\|_\infty^2] \cdot \|\mathbf{W}_{p,t}\|_F^2 + \|\mathbf{W}_{z,t}\|_F^2, \tag{19}$$

Since $\|z_t(x)\|_\infty \leq 1$, the contribution of $\mathbf{W}_{p,t}$ is strictly bounded relative to its norm. This prevents unbounded growth of effective weights even when $\|\mathbf{W}_{p,t}\|_F \to \infty$.

## B. Training Details

**Highway Network.** We implement this baseline following Srivastava et al. (2015), where each layer computes an interpolation between a non-linear transformation and a skip pathway. For any input $x$, the network computes

$$y = T(x) \odot H(x) + (1 - T(x)) \odot C(x) \tag{20}$$

where $H$ is a learnable nonlinear transformation, $T = \sigma(\text{gate}(x))$ is a sigmoid gate, and $C$ is the skip pathway. $H$ is either a linear or convolution layer, followed by a ReLU non-linear activation function. The gate $T$ is given by a parallel linear or convolution layer with its bias initialized to a negative value ($b = -1$). If input and output dimensions differ, we use a

learnable projection in the skip pathway to match shapes. We use the same encoder structure described in Section 4.1 in all network variants. Each highway layer creates a transform output, skip output, and gate values, as all pathways are fully learnable.

**ResNet-like Network.** We implement the ResNet-like baseline following He et al. (2016). Each block consists of two convolutional layers with an identity skip connection. For any input $x$, the block computes

$$y = \phi(Conv_2(Conv_1(x)) + C(x)), \tag{21}$$

where $\phi$ is a RELU non-linear activation function and $C(x)$ is the skip pathway. If channel dimensions or spatial resolution change (stride $> 1$), we use a $1 \times 1$ projection that aligns the skip dimensions with the residual one. Following He et al. (2016), we initialize the second Conv-block to zero in order to ensure that the block behaves as an approximate identity mapping.

**HyperNet-like Network.** We implement the HyperNet-like baseline as a dynamically modulated convolutional encoder. Each layer has a normal convolution and a small auxiliary network that generates input-dependent channel-wise modulation parameters. For any input $x$, the layer computes

$$y = (1 + 0.1 \tanh(\gamma(x))) \odot Conv(x) + 0.1\beta(x), \tag{22}$$

where $Conv(x)$ is the output of the base convolution, and $\gamma(x)$ and $\beta(x)$ are generated by a small hypernetwork from the pooled layer input. The hypernetwork first applies global average pooling to $x$, followed by a linear layer, and then outputs the channel-wise gain and bias terms. Thus, instead of producing full convolutional weights, the hypernetwork produces a lightweight FiLM-style modulation of the convolutional output.

**Training protocol.** The RL policy is trained using Proximal Policy Optimization (PPO) (Schulman et al., 2017). Following the framework described in (Juliani & Ash, 2024), we report the performance at the epoch level and mark task boundaries at each distribution shift. In our setup, one epoch denotes the following steps: (i) we collect buffer_size = 1024 transitions across 11 parallel environments, then (ii) perform 3 PPO passes with minibatch size set to 64. The PPO hyperparameters are set as follows: $\gamma = 0.99, \lambda = 0.95$, clip = 0.2, entropy = 0.02, learning rate = 5e-4. We train the policy for 50,000 epochs, with distribution shifts at every 5000 epochs, i.e., 5000, 10000, ..., 45000.

## C. Computational Cost Comparison

Table 2 presents the parameter counts and forward-pass FLOPs for the main architectures evaluated in this paper. We count one multiply-accumulate as a single FLOP. The conv128 encoder requires nearly 35% more computational load than the InterpConv64 variant used in our InterpLayers, despite the latter showing higher performance in later experiments.

| Encoder Variant | Params (M) | FLOPs (M) |
|---|---|---|
| Conv128 (standard) | 1.98 | 63.5 |
| Conv128 (standard+SSP+LN) | 1.98 | 67.5 |
| InterpConv64 (fullinterp) | 1.52 | 50.8 |
| InterpConv64 (convonly) | 0.99 | 49.7 |

*Table 2.* Parameter counts and forward FLOPs per inference step.

Table 3 displays the wall-clock training time and memory usage for each of the five conditions we evaluated in Fig. 2.

## D. Soft Shrink-Perturb with LayerNorm (SSP+LN)

We implement soft shrink-perturb following (Juliani & Ash, 2024), where after each optimizer step we apply the shrink and perturb update to the parameters $x$:

$$x_{\text{new}} = \alpha\, x_{\text{current}} + \beta\, x_{\text{init}}, \quad x_{\text{init}} \sim \mathcal{D}_{\text{init}}. \tag{23}$$

with $\alpha = 0.999999$ and $\beta = 0.000001$

In SSP+LN, this continuous update is combined with LayerNorm (Ba et al., 2016) applied throughout training.

| Condition | # Runs | Avg Time/Run (hrs) | Avg Final Memory (MB) |
|---|---|---|---|
| Highway | 120 | 23.65 | 8265.8 |
| ResNet-like | 120 | 21.55 | 8318.2 |
| Standard+SSP+LN | 120 | 23.71 | 8309.6 |
| Standard+Dropout | 120 | 21.68 | 8305.2 |
| InterpLayers (Ours) | 120 | 25.78 | 8311.8 |

*Table 3.* Wall-clock training costs across all experiments.

## E. ReDo

We implement ReDo following (Sokar et al., 2023). We compute the normalized activation scores for all ReLU units on the current PPO rollout buffer every 10 epochs. Those scores which fall below the threshold $\tau = 0.025$ are 'dormant'. For these dormant units, all incoming weights are reinitialized using Kaiming uniform initialization, and the corresponding outgoing weights in the next layer are set to zero.

## F. L2-init

We implement L2-init as a continuous regularization toward the initial network parameters. At initialization, we store a frozen copy of all parameters $\theta_0$. During each PPO minibatch update, we add the penalty

$$\mathcal{L}_{\text{L2-init}} = \lambda_{\text{init}} \sum_i \|\theta_i - \theta_{0,i}\|_2^2 \tag{24}$$

to the PPO loss, with $\lambda_{\text{init}} = 0.01$. The total loss is therefore

$$\mathcal{L} = \mathcal{L}_{\text{policy}} + 0.5\mathcal{L}_{\text{value}} - \beta_{\text{ent}}\mathcal{H} + \mathcal{L}_{\text{L2-init}}. \tag{25}$$

## G. Details on the Encoder-Side Gradient-Rank Computation

We measure an encoder-side gradient-rank diagnostic throughout training. The goal is to evaluate whether InterpLayers maintain gradient diversity in the part of the network where the architectural intervention is applied. This analysis is based on the gradient geometry with respect to encoder parameters, inspired by the gradient-Gram construction used in Tang et al. (2025).

**Scope.** For each model, we compute gradients with respect to a chosen encoder parameter scope. Unless stated otherwise, we use the convolutional encoder parameters. We denote the resulting diagnostic by $r_{99}^{\text{enc}}$, the encoder gradient rank at 99% spectral mass. This focuses the analysis on the learning component directly affected by InterpLayers.

**Minibatch gradient construction.** During training, we collect the PPO rollout buffer and sample a set of PPO minibatches from it before applying the PPO update. For each minibatch $b$, we compute the clipped PPO policy loss using the current model logits, the stored old log-probabilities from the rollout, and normalized advantages. We then compute the gradient of this loss with respect to the selected encoder parameters:

$$g_b = \nabla_{\theta_{\text{enc}}} \mathcal{L}_{\text{PPO}}^{(b)}.$$

Each minibatch gradient is flattened into a vector. Stacking the gradients from $B$ sampled minibatches gives

$$G = \begin{bmatrix} g_1^\top \\ g_2^\top \\ \vdots \\ g_B^\top \end{bmatrix}.$$

**Gradient-Gram matrix.** We then form the empirical gradient-Gram matrix:

$$K = GG^\top,$$

where each entry $K_{ij} = \langle g_i, g_j \rangle$ measures the alignment between PPO-loss gradients from minibatches $i$ and $j$. This matrix is not the full NTK over all individual samples and output dimensions, but an NTK-like empirical approximation in the sense that it measures local gradient geometry.

**Encoder gradient rank at 99% spectral mass.** To summarize the spectrum of $K$, we compute the singular values $\{s_i\}_{i=1}^{B}$ and sort them in descending order. We define the encoder gradient rank at 99% spectral mass as the smallest integer $k$ satisfying

$$\sum_{i=1}^{k} s_i \geq 0.99 \sum_{i=1}^{B} s_i.$$

We denote this quantity as $r_{99}^{\text{enc}}$. A higher $r_{99}^{\text{enc}}$ indicates that minibatch gradients span more independent directions, suggesting a richer and less collapsed local update geometry. A lower value indicates that most gradient energy is concentrated in only a few directions.

**Interpretation.** As we construct the Gram matrix across sampled minibatches, the maximum possible value of $r_{99}^{\text{enc}}$ is upper-bounded by the number of sampled minibatches $B$, not by the number of encoder parameters. Therefore, this metric should be interpreted as a relative diagnostic of gradient diversity across methods that use *the same* logging configuration. In our experiments, InterpLayers preserve the strongest encoder-side $r_{99}^{\text{enc}}$ among all methods, supporting our assumption that the interpolation pathway contributes additional gradient structure in the encoder.

## H. Details on the Churn Computation

We measure churn from the encoder outputs using a fixed reference batch that is stored at initialization. At epoch $t$, churn is defined as the mean squared deviation of the current encoder representations from the initial ones:

$$\mathcal{C}_t = \mathbb{E}_{x \sim \mathcal{D}} \left[ \|h_t(x) - h_0(x)\|_2^2 \right], \tag{26}$$

where $h_t(x)$ denotes the encoder representation of input $x$ at epoch $t$, and $h_0(x)$ the corresponding representation at initialization. We also log the first- and second-order finite differences of $C_t$ over epochs.

## I. Visualization of the Distribution Shifts of ProcGen Tasks

Sample visualizations for three ProcGen coinrun tasks evaluated in this paper are shown in Figure 7. For **permute**, a fixed random pixel permutation is applied per shift. Given the change in the entire state space, this task requires robust feature relearning. For **window**, the environment is resampled with a different random seed to create other environments. Finally, the expand tasks increase the number of training environments from 100 to 1000 across 9 shifts. This characteristic evaluates the generalization capabilities of the trained policy.

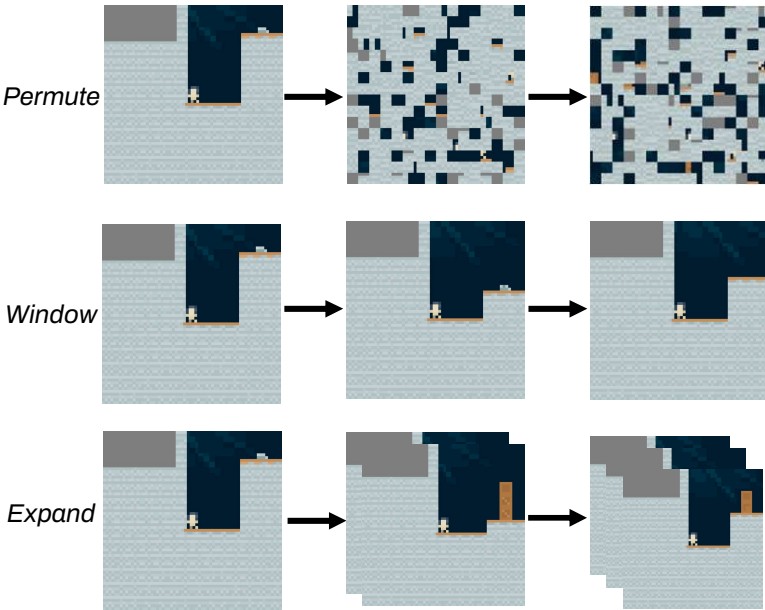

*Figure 7.* **Visualization of the distribution shifts used in ProcGen coinrun.** Each panel shows the transformation applied at the shift boundaries (every 5,000 epochs): **Permute** applies a fixed random pixel permutation per shift; **Window** resamples the environment seed to generate new levels; **Expand** increases the number of training environments from 100 to 1000 across 9 shifts. Visual representations of the environments are shown.

## J. Extended Discussion

### J.1. Why dropout improves InterpLayer results

Different from the standard application in neural networks, where dropout masks neurons in all layers, in our method, we only apply dropout to the projection pathway. Our intuition is that applying dropout only to the projection increases the activation variance of the projection pathway, affecting the representational gap $D$ defined in Theorem 3.1. In Theorem 3.1, we show that InterpLayers provide an upper bound to their output variability by:

$$\|\Delta h(x)\|_2 \;\le\; \|z(x)\|_\infty \, L_p \, \|\Delta\theta_p\|_2 \;+\; L_z \, \|\Delta\theta_z\|_2 \, D(x), \qquad D(x) = \|h_{\mathrm{proj}}(x) - h_{\mathrm{ref}}(x)\|_2. \tag{27}$$

In this equation, the term $D$ determines how much changes in the interpolation weights affect the output. In this case, if the projection and reference are equal, the interpolation weights do not have an impact on the output. However, if $D$ increases, the impact of the interpolation pathway also increases. Dropout strictly increases the activation variance in the projection pathway by injecting noise. So the instantaneous gap is as follows

$$D_t(x) = \left\|\tilde{h}_{\mathrm{proj},t}(x) - h_{\mathrm{ref}}(x)\right\|. \tag{28}$$

This way, dropout guarantees that $D_t(x)$ always has some variance, stabilizing the gradient of the interpolation pathway $z(x)$ in Equations 4 and 7.

**InterpLayers with Dropout vs Standard Networks with Dropout.** In Figure 2, we compare InterpLayers to standard networks with dropout. We show that applying dropout also improves the plasticity of standard networks significantly. However, it does not fully prevent plasticity loss across all conditions, and they still perform worse than InterpLayers. We hypothesize that this difference in performance happens because standard layers do not benefit from the additional activation variance as InterpLayers, which contain both a reference and a projection pathway. Instead, for standard networks, dropout decreases model-level variance, which slows down the performance collapse but does not prevent it, as the empirical results suggest.

### J.2. Plasticity Index

Following the intuition from the previous section, we list two independent mechanisms that control plasticity in InterpLayers:

1. The representational gap $D$

$$D(x) = \|h_{\text{proj}}(x) - h_{\text{ref}}(x)\|_2.$$

   High values for $D$ mean that even small changes to the interpolation weights yield large changes in the output.

2. The interpolation weights $z$
   If these weights are close to 0 (or 1, for that matter), the magnitude of $D$ becomes less impactful on the final output.

When combined, $z$ can be interpreted as a "exposure" term to the projection pathway and $D$ as a "sensitivity" term to that exposure. Combining them quantifies how "plastic" a layer is, which we define as a plasticity index:

$$PI(x) = z(x)\, D(x).$$

If $PI = 0$, the layer shows no plasticity because either the projection has no influence on the output (if $z = 0$) or because the reference and projection are indistinguishable (if $D = 0$). $PI > 0$ means that the layer is somewhat plastic because the projection is different from the reference, and the interpolation exposes that difference. Dropout affects this index due to its "variance injection" (if we want to follow the common term of plasticity injection). This variance ensures that even if $D$ is overall decreasing, the instantaneous $D_t$ will show some fluctuations. As the interpolation weights' gradients are dependent on $D$, this means that they would never become dormant. This maintains $PI$ to be non-zero.

### J.3. Interpolation distributions maintain variance

Figure 6 show that the interpolation weights do settle around 0.2 across all tasks. However, this illustration alone does not provide enough information about whether they maintain their variance. To evaluate their variance, we compute the Normalized Gate Diversity Ratio (NGDR) for each layer

$$NGDR(t) = \frac{Var[z_t]}{\mu_t(1 - \mu_t)} \tag{29}$$

where $z_t$ denotes the interpolation weights of a layer at epoch $t$ and $\mu_t$ is their mean. The denominator is the variance of a Bernoulli distribution with mean $\mu_t$, so that the NGDR works as a scale-free measure of how diverse the interpolation weights are relative to the maximal possible variance. Figure 9 shows that across all shifts in all tasks, the interpolation values saturate toward values around 0.2 (as mentioned above), whereas their NGDR remains stable (between 0.3 and 0.5) during training. This suggests that the interpolation weights do not collapse to a single value but instead remain or even increase their variance.

## K. Guidelines for choosing InterpLayer variants

InterpLayers can be applied to any layer in a neural network. In our ProcGen environment, the network architecture is an encoder followed by respective PPO heads, following (Juliani & Ash, 2024). The encoder consists of four convolutional layers and one linear layer. The convolutional stack acts as a feature encoder, with the linear layers combining these features accordingly. In this work, we evaluate two InterpLayer variants:

$$\textbf{convonly} : \text{InterpLayer is only applied to the convolutional layers of the encoder.} \tag{30}$$
$$\textbf{fullinterp} : \text{InterpLayer is applied to all layers of the encoder.} \tag{31}$$

Omitting the InterpLayer from the linear layer reduces the parameter count by 524,544. This difference is significant if memory or throughput are limiting factors. Across all three task shifts, the convonly variant performs equal to or better than the fullinterp variant, especially when combined with dropout. This suggests that most of the benefit of interpolation occurs in the convolutional part of the encoder. Thus, we recommend using the **convonly** variant when the architecture has a clear feature encoder or computational efficiency is important, and to use **fullinterp** in scenarios where there are no computational restrictions or the training is unstable after task changes.

## L. Extended results

### L.1. Raw results

We show the raw returns graph obtained during training in Fig. 8. It is seen that SSP+LN consistently achieves the highest rewards. InterpLayers achieve good performance, especially for the window and expand tasks. It is interesting to observe

that, even though Highway Networks show plasticity loss in different scenarios, they achieve convergence speed and raw reward similar to SSP+LN for the first tasks.

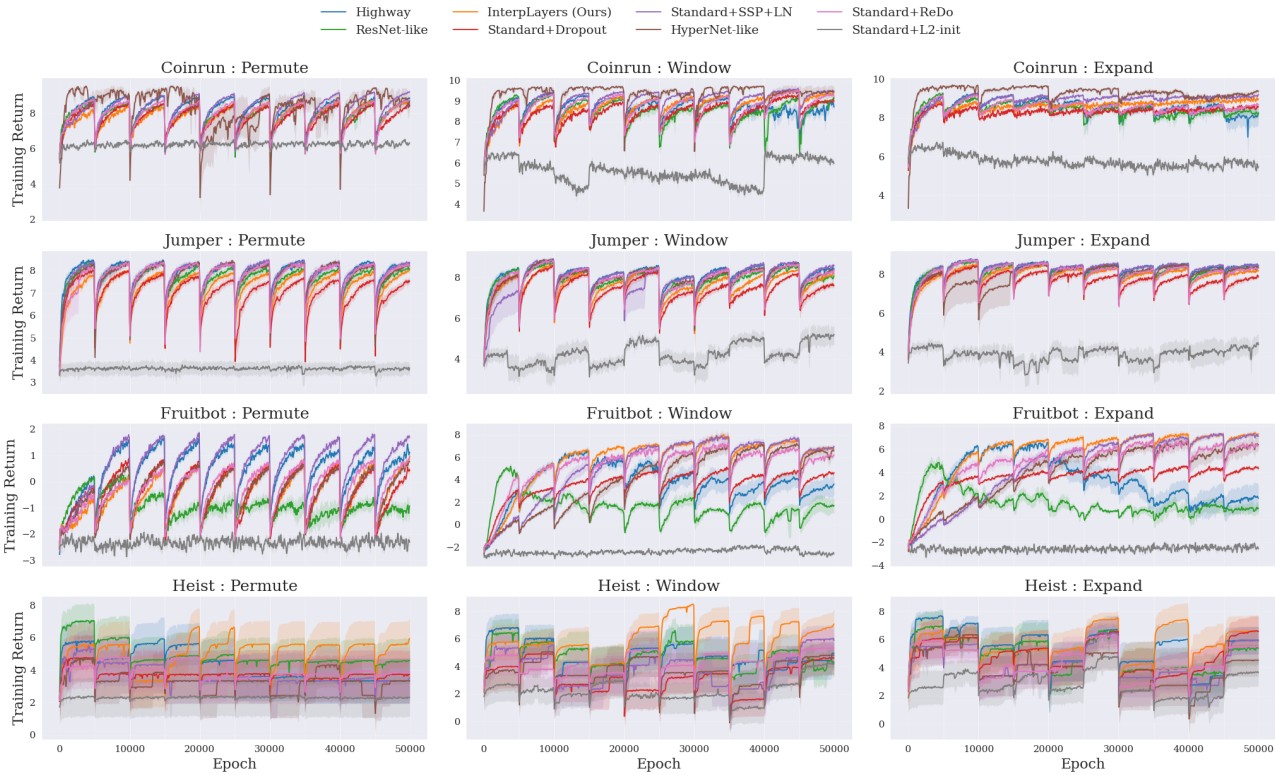

*Figure 8.* **Raw rewards obtained during training for the methods evaluated.**

### L.2. Influence of Dropout Rate

We conduct a small ablation study in the Coinrun environment using different dropout rates ranging from no dropout to a dropout rate of 0.2 for 6 seeds each. Figure 10 shows the performance retention and raw returns of different variants for the three distribution shifts. It is observed that higher dropout rates lead to minimally higher relative performance, however, at the cost of losing raw reward performance. We can also observe that no dropout, while having very high initial performance, shows strong performance degradation, especially in the expand condition. Therefore, it is important to choose a dropout rate that keeps plasticity while being still able to solve the environments where the policy will be applied. In this work, we choose a dropout rate of 0.05 for the default InterpLayer variant.

### L.3. NTK Analysis

In Theorem 3.2, we outline theoretically how InterpLayers have properties that prevent the NTK rank from collapsing. Here, we measure encoder-side gradient-rank diagnostics following the methodology described in Appendix G. Fig. 11 shows the encoder-side $r_{99}^{enc}$ of InterpLayers and the baselines in the Coinrun environment. The results show that InterpLayers maintain high encoder-side rank throughout training, while several baselines show stronger rank reduction in late training. This analysis is still limited in scope, and future work should explore whether these encoder-side gradient-rank patterns continue to correlate with mitigating plasticity loss during extended training periods in continual learning settings.

## M. Comparison of InterpLayers variants

We evaluate two architectural variants: (i) **convonly**, where InterpLayers replace only the convolutional encoder layers, and (ii) **fullinterp**, where both convolutional and linear layers are replaced with InterpLayers. We also investigate InterpLayers combined with dropout (Srivastava et al., 2014), which we name **convonly-dropout** and **fullinterp-dropout** respectively, in

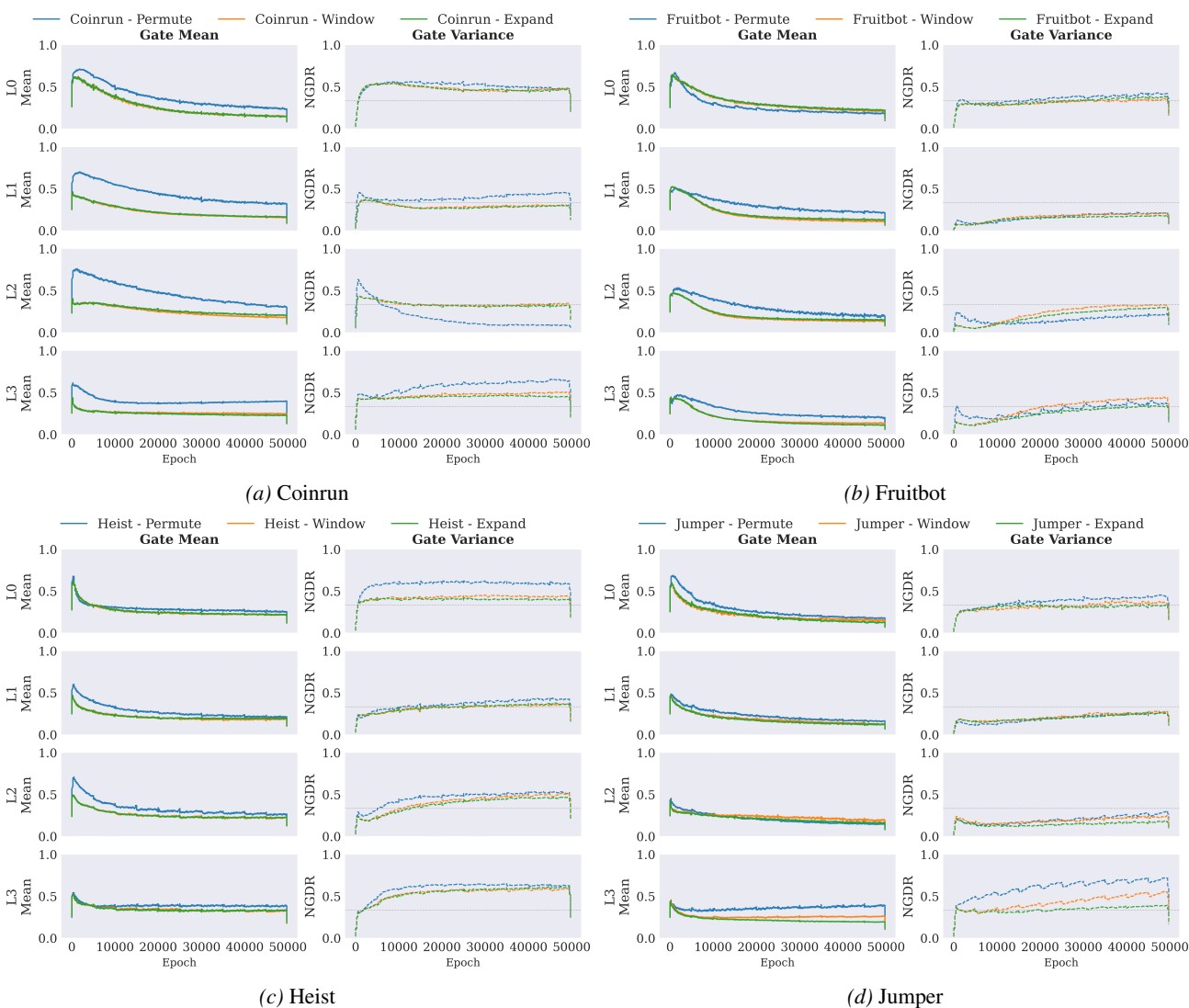

*(a)* Coinrun

*(b)* Fruitbot

*(c)* Heist

*(d)* Jumper

*Figure 9.* **Evolution of the layer-wise gate means and NGDR across tasks and shift types.** The interpolation means settle around $\approx 0.2$, while the NGDR remains stable between 0.3 and 0.5 even in late training stages. This indicates that the gate variance does not collapse which supports the theoretical claim in Theorem 3.2 that InterpLayers maintain gradient diversity thus a stable NTK rank.

which dropout is applied to the projection pathway. The convonly variant emphasizes stability in low-level feature extraction, while fullinterp exposes the entire network to interpolation.

We compare InterpLayer variants in the Coinrun environment using 10 random seeds for each condition in each shift. Figure 12 shows that the non-dropout variants outperform the dropout ones in terms of raw performance in the permute condition, but they show significant performance degradation in the other two tasks. Between convonly-dropout and fullinterp-dropout variants the performance is similar. We chose the **convonly-dropout** as our default variant throughout our experiments as it is computationally cheaper than the fullinterp-dropout variant (see Section C).

## N. Ablation with Permuted Task Order

In this section, we aim to verify that InterpLayers are not overfitting on a specific task sequence, i.e., that they learn the pattern of a shift and not actually the newly presented task. For this, we performed all shifts a priori and stored the environments. Then we shuffled the task sequence so that an original sequence of 1-2-...-9 would become, for example, 1-5-...-3. Figure 13 shows that InterpLayers reach the same level of performance retention and raw performance in both

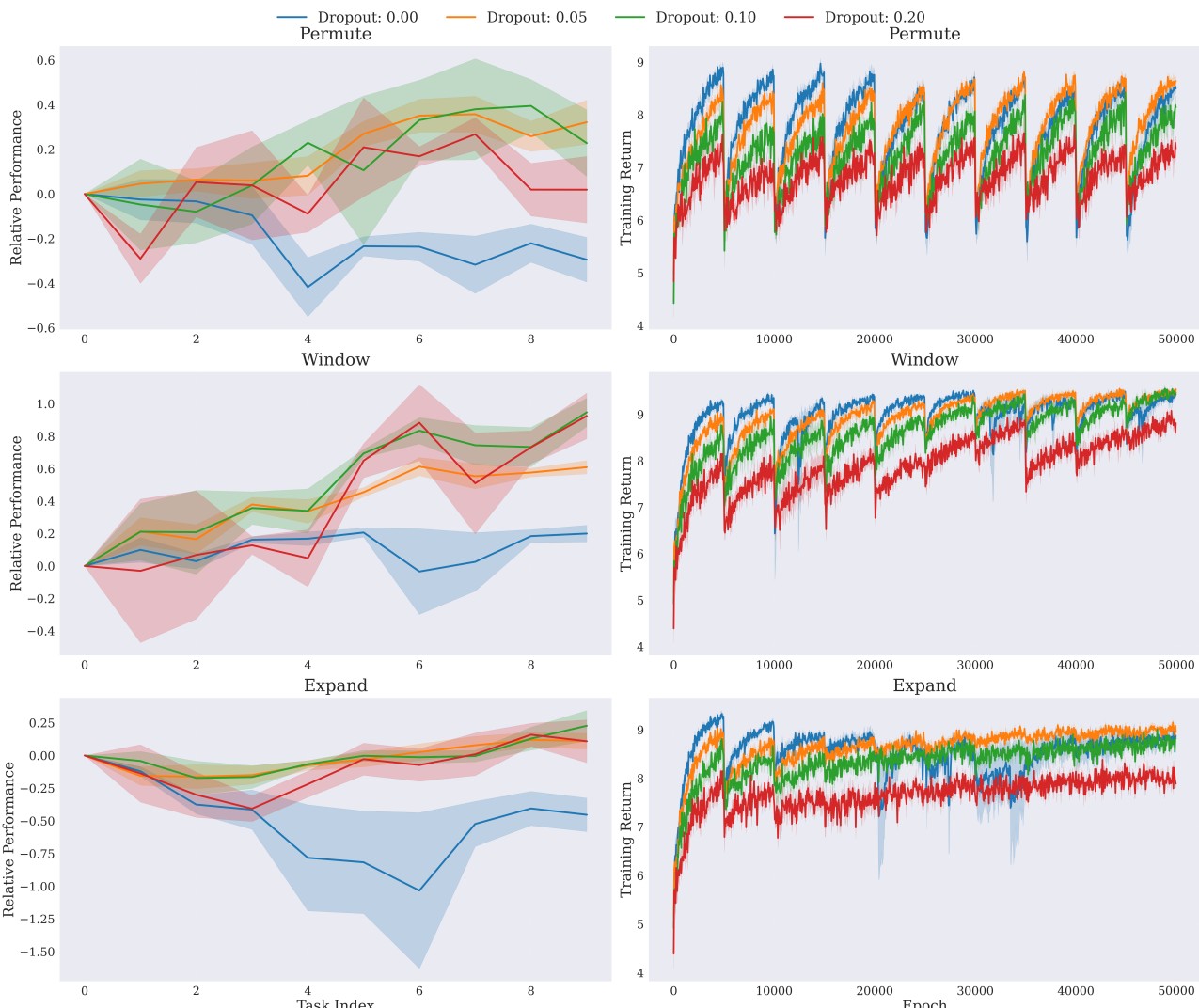

*Figure 10.* **Ablation study for the dropout rates applied to the projection-pathway.** Higher dropout rates show minimally better relative performance but show strong decreases in raw returns, while removing dropout (0.00) shows substantial performance degradation, especially in the expand shift. The dropout rate of 0.05 shows the best balance between performance retention and raw performance.

settings (original and permuted). In this way, we show that InterpLayers are robust to random task sequences, an important property in continual learning settings.

## O. Ablation of initialization of interpolation weights

In both (Qiu et al., 2025) and our study, we have observed that the mean of the gate distributions converges to values between 0.1 and 0.3. This brings up the question of whether this is caused by initializing them with a mean at 0.5. Therefore, we ran two ablation studies where we initialized the interpolation weights with a mean at 0.25 and 0.75, respectively. Firstly, Figure 14 shows that the different initialization of the parameters does not affect the performance of InterpLayers. Secondly, Figure 15 depicts that in both cases, 0.75 and 0.25, the interpolation means show a similar pattern as when initialized at 0.5 in the window and expand condition. At the beginning of the training there is a small rise of the mean that is followed by a decrease and then convergence to a mean of around 0.2 across all layers, with the standard initialization having a slightly lower mean in the last layer. In the permute condition, we can see a similar pattern between the conditions. However, for both ablations, the mean converges to a much larger value of around 0.4 in the deeper layers whereas the standard initialization converges to 0.25.

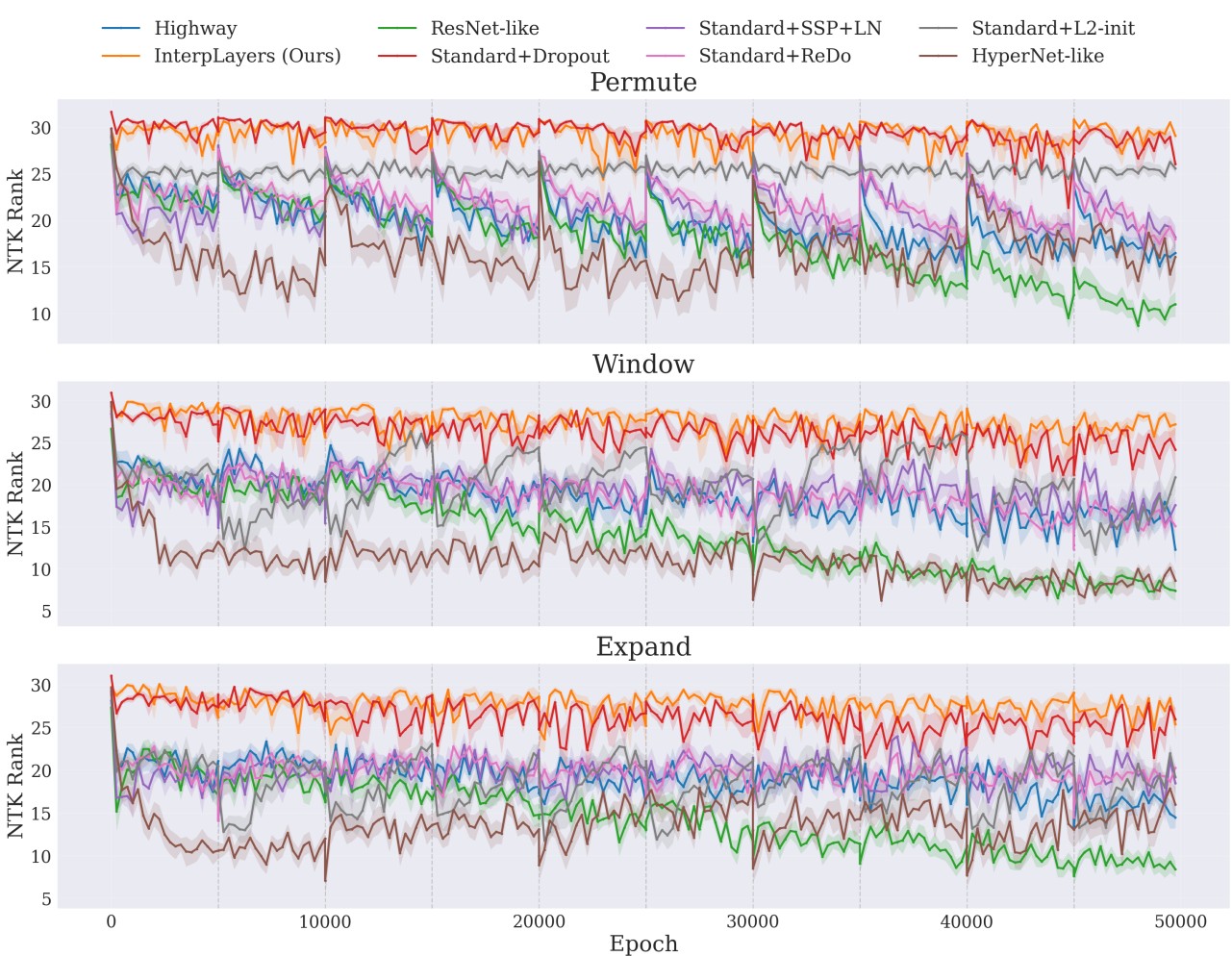

*Figure 11.* **Evolution of effective NTK rank for InterpLayers and baselines under distribution shifts in Coinrun.** Shaded regions denote variability across 10 seeds, and vertical lines indicate shift points. Across all conditions, all variants show stable rank besides SSP+LN which first shows a strong increase followed by a gentle decrease.

## P. Continual Gym Control Environment

In this section, we test whether InterpLayers prevent plasticity loss in environments beyond ProcGen. We run additional experiments on continual Gym Control following the setup of Tang et al. (2025), using the same environments, task counts, and perturbation scales. We evaluate four environments: C-Acrobot, C-CartPole, C-LunarLander, and C-MountainCar.

To ensure a fair comparison between methods, we only tune hyperparameters shared across all methods, namely the learning rate, entropy coefficient, and number of PPO update passes. We use nine seeds for each setting. Following the evaluation setup used by Tang et al. (2025), we compute the final performance $J(T)$ for each method and environment, and then average these values across environments to obtain the aggregate score.

For method $m$ and environment $e$, let $J_e^{(m)}(T)$ denote the final task-sequence performance after the full learning horizon. The aggregate score is computed as

$$J_{\text{agg}}^{(m)} = \frac{1}{|\mathcal{E}|} \sum_{e \in \mathcal{E}} J_e^{(m)}(T),$$

where $\mathcal{E}$ denotes the set of Gym Control environments. Higher values indicate better performance.

Table 4 shows that InterpLayers retain their performance beyond ProcGen and achieve comparable results to the strongest baselines. SSP+LN achieves the highest aggregate score with $-94.490$, followed by InterpLayers with $-97.101$. InterpLay-

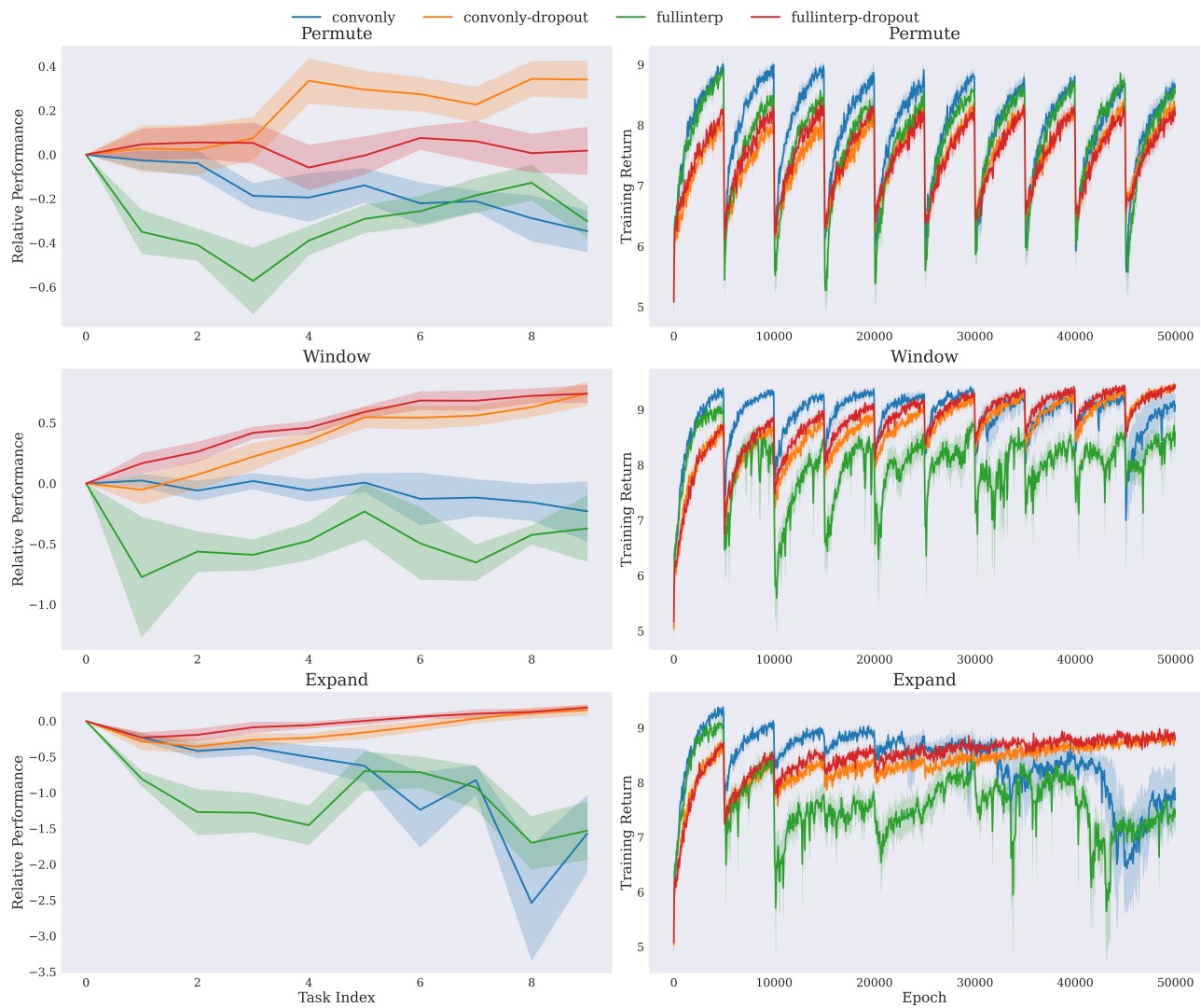

*Figure 12.* **Comparison of different InterpLayer variants in Coinrun.**

ers are strongest in C-CartPole and C-MountainCar, and second-best in C-LunarLander. SSP+LN is best in C-LunarLander and second-best on C-CartPole and C-MountainCar. These results show that InterpLayers are on par with the best algorithmic methods for preventing plasticity loss while using an architectural mechanism for performance retention.

## Q. Ablation of Optimizer

In this section, we aim to verify that the ability of InterpLayers to retain performance is insensitive to a specific optimizer choice. For this, we performed an ablation in CoinRun with four different optimizers: Adam, AdamW, RMSProp, and SGD. Fig. 16 shows that InterpLayers reach stable performance retention and raw performance for all optimizers except SGD. The Adam variants achieve slightly stronger raw performance than RMSProp. For SGD, we observe a large performance decrease. However, we observe an even stronger performance decrease when using Standard+Dropout with SGD, suggesting that this degradation is not specific to InterpLayers but a general limitation of this optimizer in this setting. The results indicate that InterpLayers are robust to a range of commonly used optimizers, especially with respect to performance retention.

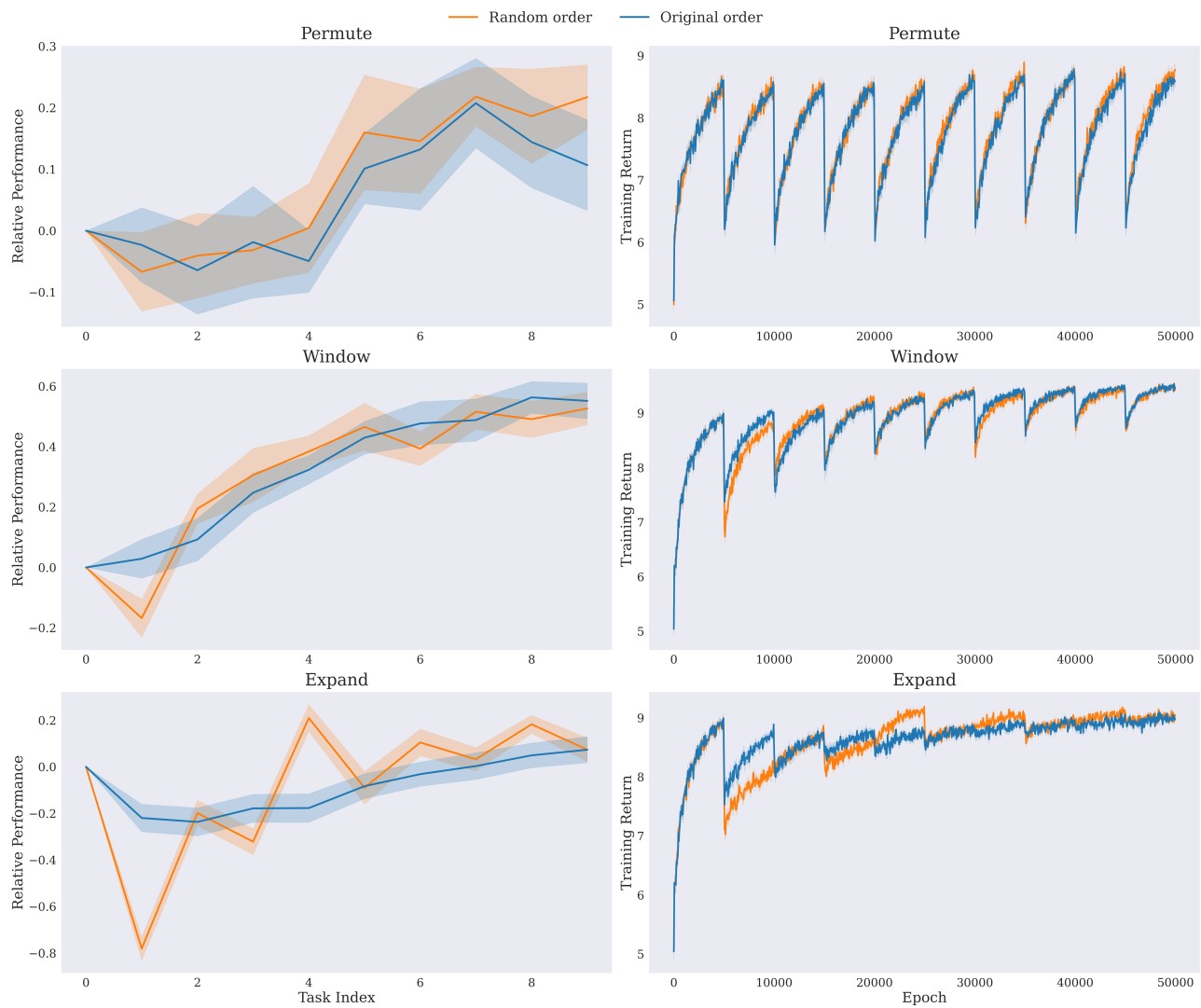

*Figure 13.* **Results for ablation study with task order permuted.** InterpLayers show consistent results for both settings, showing robustness to randomly permuted tasks.

## R. Ablation of Learning Rate

In this section, our aim is to show that InterpLayers are robust to changes in learning rates. Here, we perform an ablation on CoinRun with learning rates $\eta \in \{5e{-}5, 1e{-}4, 5e{-}4\}$. Fig. 17 shows that InterpLayers have stable performance retention for $\eta = 5e{-}5$ and $\eta = 1e{-}4$, whereas $\eta = 5e{-}4$ shows slightly higher variance and a strong drop in the penultimate task in Permute and along the later shifts in Expand. The learning rate $\eta = 1e{-}4$ achieves the best raw performance and is on par with $\eta = 5e{-}5$ in terms of performance retention. These results indicate that InterpLayers are robust to the learning rate in our setting, especially for $\eta = 5e{-}5$ and $\eta = 1e{-}4$.

## S. Quantifying Plasticity Loss

There is no standardized metric to quantify network adaptability in continual reinforcement learning. In our experiments, we therefore make use of a common choice among recent papers: relative performance. We define plasticity loss as the relative decrease in an agent's ability to learn later tasks compared to the initial task. Let $J_i$ denote the mean performance on task $i$, computed over the final $K$ training epochs for that task. In our experiments, we use $K = 50$. We define relative

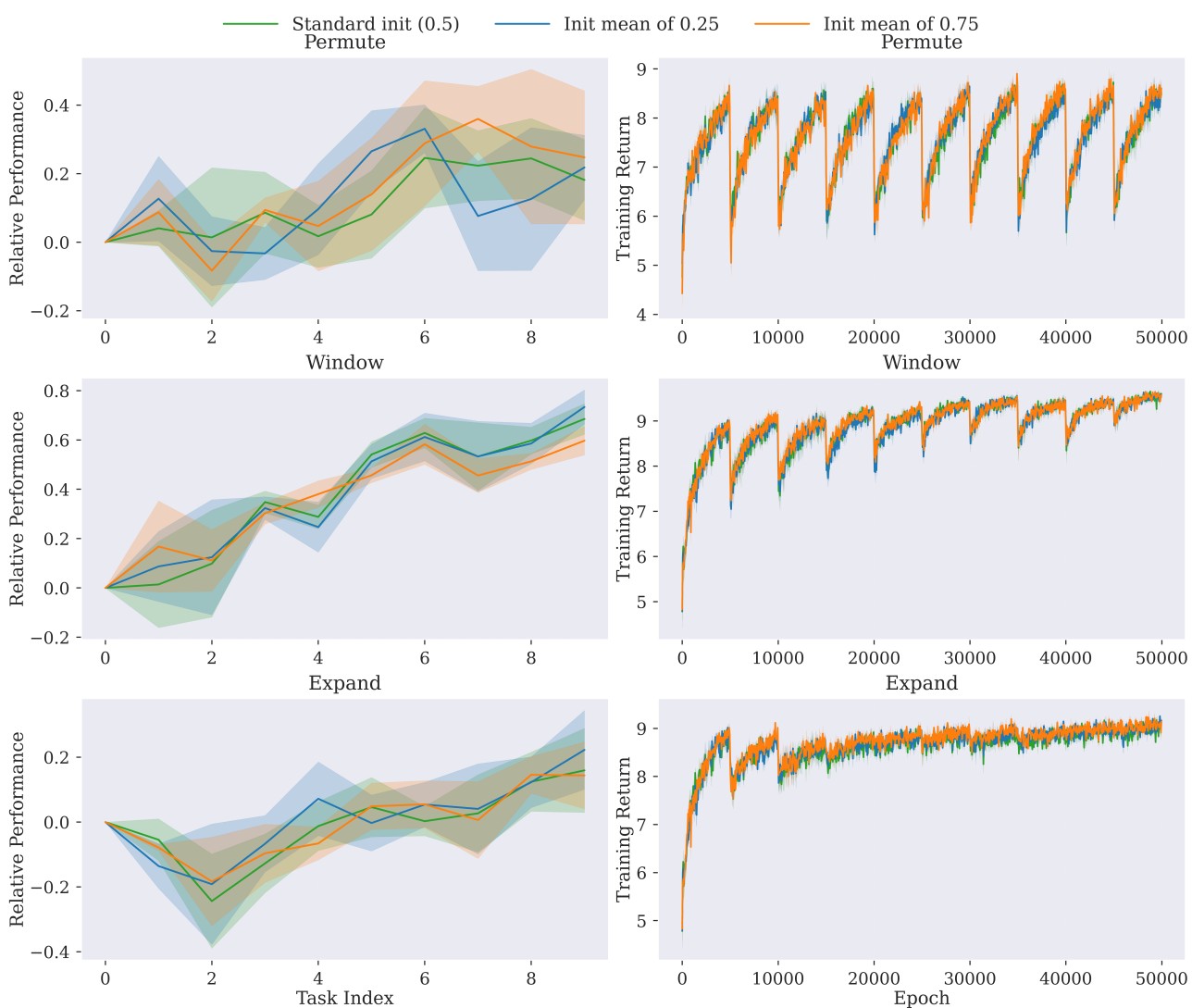

*Figure 14.* **Performance results of the ablation study for interpolation weight initialization values.** Both ablations, initialization with a mean of 0.25, and 0.75, show similar performance as the standard initialization with a mean of 0.5.

performance as

$$\Delta J_i = J_i - J_0, \tag{32}$$

where $J_0$ is the final performance on the initial task for the same run. Thus, $\Delta J_0 = 0$ by construction. Negative values indicate that the agent reaches lower performance on later tasks than on the initial task, indicating a decrease in plasticity. We compute $\Delta J_i$ separately for each seed and then report the mean and standard error across seeds.

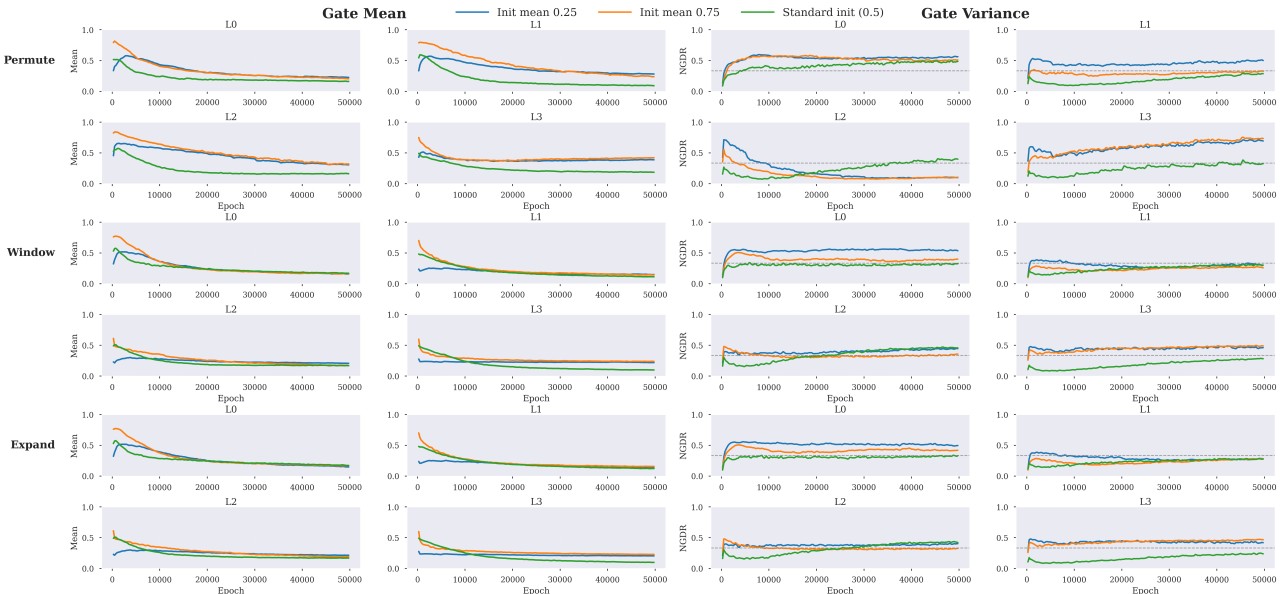

*Figure 15.* **Mean and variance in the ablation study for interpolation weight initialization values.**

*Table 4.* Performance comparison on continual Gym Control. Scores are best tuned $J(T)$ values, shown as mean $\pm$ SE over nine seeds. Best values are bold and underlined; second-best values are underlined.

| Env | Standard | Standard + Dropout | Standard + SSP+LN | L2-init | InterpLayer (Ours) | HyperNet-like |
|---|---|---|---|---|---|---|
| C-Acrobot | $-139.974 \pm 11.753$ | $-110.181 \pm 6.741$ | $\underline{-104.588 \pm 3.318}$ | $\mathbf{-101.415 \pm 1.376}$ | $-105.214 \pm 5.033$ | $-116.001 \pm 7.469$ |
| C-CartPole | $45.349 \pm 7.989$ | $49.139 \pm 6.725$ | $\underline{54.204 \pm 6.179}$ | $29.804 \pm 2.071$ | $\mathbf{56.360 \pm 11.889}$ | $49.797 \pm 8.043$ |
| C-LunarLander | $-215.899 \pm 13.793$ | $-177.832 \pm 12.194$ | $\mathbf{-153.874 \pm 10.355}$ | $-208.060 \pm 5.956$ | $\underline{-174.383 \pm 14.574}$ | $-206.567 \pm 24.121$ |
| C-MountainCar | $-188.802 \pm 4.484$ | $-183.003 \pm 5.261$ | $\underline{-173.703 \pm 6.249}$ | $-200.000 \pm 0.000$ | $\mathbf{-165.166 \pm 3.063}$ | $-178.195 \pm 3.142$ |
| Agg. Score | $-124.831$ | $-105.469$ | $\mathbf{-94.490}$ | $-119.918$ | $\underline{-97.101}$ | $-112.742$ |

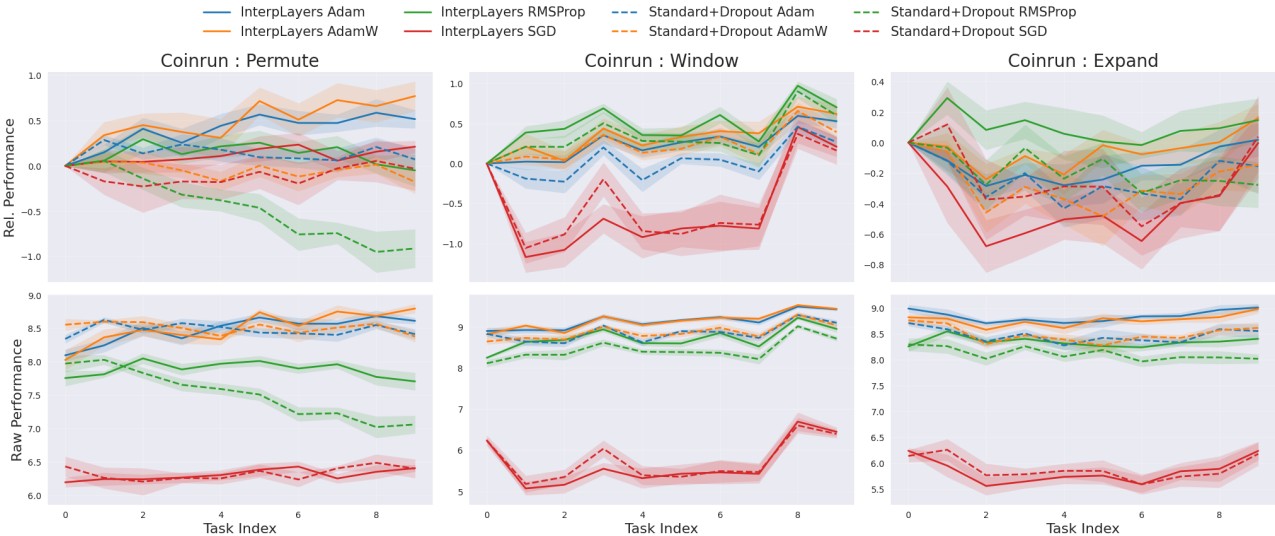

*Figure 16.* **Performance results of the optimizer ablation study.** InterpLayers show stable raw performance and performance retention for Adam, AdamW, and RMSProp. With SGD, both InterpLayers and Standard+Dropout show a performance decrease, suggesting that the degradation is due to the optimizer rather than the InterpLayer mechanism.

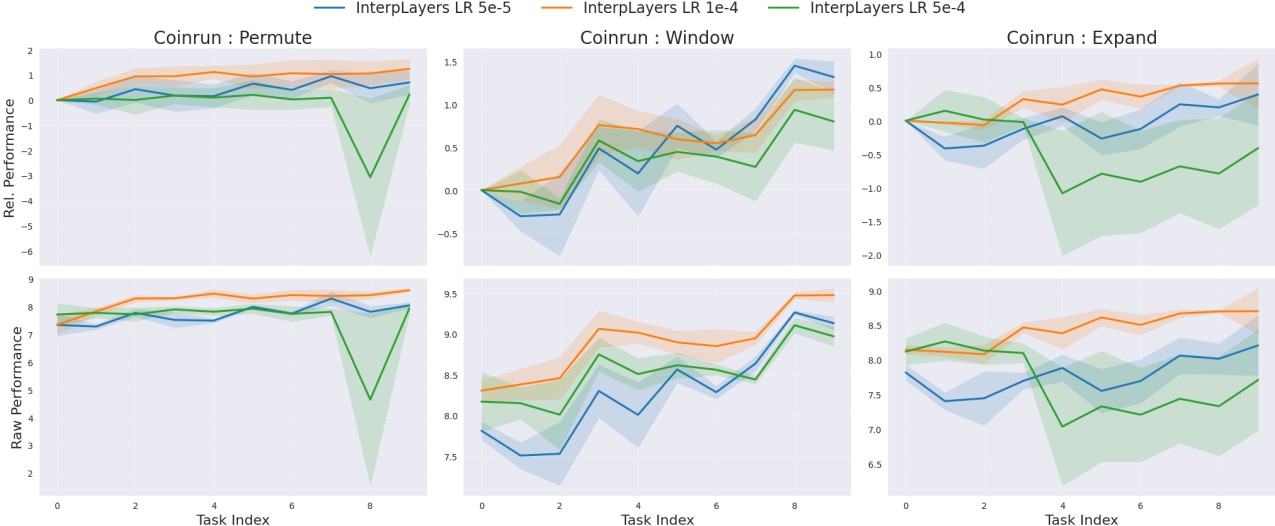

*Figure 17.* **Performance results of the learning-rate ablation study.** InterpLayers show stable performance retention across learning rates $\eta \in \{5e-5, 1e-4, 5e-4\}$, with $\eta = 1e-4$ achieving the strongest overall performance.

