# OpenReview forum: "Mitigating Plasticity Loss through Architectural Design in Continual Learning"
_ICML.cc/2026/Conference — ICML 2026 regular_

### Official Review · Reviewer_9DoX · 2026-02-26

**Soundness:** 2
**Presentation:** 3
**Significance:** 3
**Originality:** 2
**Overall Recommendation:** 4
**Confidence:** 5

**Summary:**

This paper introduces InterpLayer, an architectural mechanism designed to mitigate plasticity loss in continual reinforcement learning (CRL). The method enhances standard network layers by incorporating a fixed reference pathway and a learnable projection pathway, which are combined using input-dependent interpolation weights. The authors provide theoretical insights related to representational drift and Neural Tangent Kernel (NTK) rank collapse, and evaluate the approach on ProcGen environments under various types of distributional shifts. Overall, the work presents a simple architectural alternative to algorithmic methods for preserving plasticity in CRL.

**Compliance With Llm Reviewing Policy:**

Affirmed.

**Final Justification:**

The authors solved my concerns.

**Key Questions For Authors:**

1) How does this method perform on standard continual RL benchmarks (e.g., Atari, MuJoCo)? Tests on standard benchmarks will help to assess generalizability and enable broader comparisons.
2) Considering the architectural resemblance between InterpLayers and residual layers, it is important for the authors to provide a clear justification as to whether the reported improvements are attributable to the interpolation design itself, rather than to general architectural changes.
3) Empirically, preserving NTK rank should correlate with improved continual learning performance. Do the results presented in the paper support this correlation?

**Limitations:**

No limitation discussion

**Strengths And Weaknesses:**

Strengths

1) Plasticity loss is addressed architecturally rather than through optimization, enabling a straightforward, lightweight solution.
2) The paper provides theoretical motivation based on representational stability and NTK analysis.
3) Experiments cover multiple types of distributional shifts within ProcGen.

Weaknesses

1) The experiments are restricted to ProcGen environments, using CoinRun, Jumper, Fruitbot, and Heist with three types of distribution shifts (permute, window, and expand). While these shifts introduced some variation within each game, they largely remain single-task variants with relatively simple 2D dynamics. As a result, the evaluation may not fully assess the model’s plasticity, resistance to forgetting, or long-term adaptation across truly diverse or sequential tasks, which are central challenges in continual reinforcement learning. In particular, the permute shift disrupts pixel structure but does not introduce new task objectives, and the expand/window shifts mainly increase level variety rather than heterogeneity of tasks, limiting insight into continual learning in more complex or heterogeneous environments.
2) Although the method is presented as an architectural solution, the evaluation is limited to gated variants and algorithmic baselines. Other structure-level mechanisms discussed in the literature: such as skip and residual connections, input-dependent gating, and HyperNetworks, are not included in the comparison. Without such comparisons, it remains difficult to determine whether the observed improvements arise specifically from the proposed interpolation design or from more general architectural modifications.
3) The paper argues that InterpLayer mitigates NTK rank collapse and stabilize representations. However, the empirical link between NTK rank dynamics and performance is not clearly established. It remains unclear whether preserving NTK rank consistently correlates with improved continual performance, leaving the theoretical claims somewhat detached from observed results.
4) The interpolation weights consistently converged to a stable value, but the implications of this behavior are not deeply analyzed (e.g., task dependence or sensitivity). In addition, while parameter counts are discussed, the paper lacks a detailed characterization of runtime and scalability, making it difficult to assess practical trade-offs.

---

> ### Author Rebuttal · Authors · 2026-03-31
>
> The authors are thankful to the reviewer for your insightful and detailed comments.
>
> ---
> ### W1/Q1. (Additional Benchmarks)
>
> Thank you for this suggestion. To assess the generalizability of InterpLayers beyond ProcGen, we ran additional experiments on continual Gym Control following the setup of Tang et al. (2025; Appendix A.1), using the same environments, task counts, and perturbation scales.
>
> For fair comparisons, we tuned only hyperparameters shared across all methods: learning rate, entropy coefficient, and number of PPO update passes. Following Tang et al.'s aggregated-score evaluation metric, we averaged performance across the full continual task sequence within each environment and then across environments running 6 seeds for each setting.
>
> Under this metric, SSP+LN scores -93.517, followed by InterpLayers+Dropout with -94.506, a non-significant difference. If we review the individual tasks InterpLayers outperform SSP+LN in Acrobot (-98.900 vs. -104.651), CartPole (64.215 vs. 55.511), and MountainCar (-168.334 vs. -169.526) while SSP+LN achieves better performance in LunarLander (-155.403 vs. -175.004). The results for other baselines are Standard+Dropout (-108.262), Hypernet (-116.435), L2 Init (-121.455), and Standard (-125.707). These results show that InterpLayers generalize beyond ProcGen and remain competitive with the strongest baselines.
>
> ---
> ### W2. (Baseline Methods)
>
> In our work, we compared InterpLayers against two architectural baselines: ResNet-like (fixed skip connection) and Highway (input-dependent gating). Following the reviewer's suggestion, we add a HyperNetwork-style [1] baseline during the rebuttal. In CoinRun, the HyperNetwork baseline performs well on window and expand but is unstable on permute. In Gym Control, its results are weaker when compared to InterpLayers and SSP+LN. We will revise the manuscript and make the distinction between residual, gating, hypernetwork, and interpolation-based architectures clearer. Interestingly, hypernetwork-style modulation appears orthogonal to InterpLayers opening future research opportunities.
>
> [1] Ha, David, Andrew Dai, and Quoc V. Le. "Hypernetworks." arXiv:1609.09106 (2016).
>
> ---
> ### W3/Q3. (NTK Results)
>
> Thank you for raising this important point. To better assess whether stronger NTK rank preservation is reflected in stronger continual performance, we refined the analysis to measure NTK rank in the encoder, where InterpLayers act most directly, following Tang et al. (2025) and summarizing the spectrum with the 99\% mass approximate rank. In this new analysis, the stronger-performing variants also show more preserved encoder-side rank, while weaker baselines such as Standard or ResNet-like show much lower rank. This makes the empirical evidence more consistent with our theoretical results. See the response to W2/Q2 of reviewer 5AjA for details regarding the methodology and results for this new analysis.
>
> ---
> ### W4. (Additional Analysis)
>
> According to the reviewer's suggestion, we analyze further the interpolation weights' patterns. Specifically, our observation is that the layer-wise mean of interpolation weights stabilizes, often around $z\approx0.2$, while maintaining strong variance (NGDR remains around $0.3$--$0.5$). This maintenance  of strong variance is important, as a collapse to a single value would weaken the non-degenerate interpolation features needed for gradient diversity, whereas a stable mean, with preserved variance, is compatible with the NTK theorem. We will revise the manuscript to make this interpretation more explicit.
>
> Regarding the practical trade-offs, InterpLayers increase wall-clock training time by at most 10\% in our setup, with no increase in memory usage. This analysis is shown in Table 2 and Table 3 of our manuscript, and we will direct to these more clearly.
>
> ---
> ### Q2. (Interpolation and Residual Layers)
>
> The main difference between InterpLayers and residual layers is that InterpLayers introduces a learnable convex interpolator between the fixed reference pathway and the learnable pathway. By contrast, a residual layer uses a fixed additive skip connection, so the skip and transformed pathway are *always* combined the same way. To evaluate the effects of the learnable interpolation design, we have a residual layer baseline (ResNet-like) that uses a fixed additive skip connection. In the experimental results (Fig. 2), we observe that performance is greatly improved by the learnable formulation. We also observe that the average values for the interpolation weights is around 0.2 (Fig. 6). These results support our claim that these improvements arise from the interpolation mechanism rather than from a general architectural modification. We will clarify the differences between InterpLayers and residual layers and justify the robustness of our experimental results.
>
> ---
> We believe to have addressed all reviewer's concerns. If there are any further questions or concerns we will be happy to address them.

---

> > ### Author Rebuttal · Reviewer_9DoX · 2026-04-01
> >
> > I think the authors solved my questions. I will raise the score to 4.

---

> > > ### Author Response · Authors · 2026-04-03
> > >
> > > Thank you for acknowledging our rebuttals! Your comments helped us to broaden our empirical scope, to improve our architectural comparison, and to improve the connection between our theoretical motivation and empirical evidence!

---

### Official Review · Reviewer_5AjA · 2026-03-11

**Soundness:** 3
**Presentation:** 3
**Significance:** 3
**Originality:** 3
**Overall Recommendation:** 5
**Confidence:** 4

**Summary:**

The paper proposes to mitigate plasticity loss through an architectural intervention. Namely, they propose to modify usual linear layers (like mlp or convolution) and replace them to let the network choose between going through the layer or simply going through the identity. The choice is implemented through an interpolation mechanism that can also be learnt. The authors argue that such mechanism may prevent churning and neural tangent kernel rank collapse.
The authors present experiments on several continual reinforcement learning tasks (three environments, and three task variations). They compare their proposed architectural intervention to algorithmic interventions. They first do a quick ablation on their proposed approach to select the most promising one, adding dropout to their approach. The proposed approach appears to perform generally on par with the current state of the art algorithmic intervention. But as the authors point out, both approaches can be combined as they act on orthogonal training components. The authors also show that their approach efficiently avoid churning compared to other architectures. The capacity of the approach to avoid neural tangent kernel rank collapse is less clear.

**Compliance With Llm Reviewing Policy:**

Affirmed.

**Final Justification:**

The rebuttal fully addressed my main concerns.

**Key Questions For Authors:**

- Can the authors have an ablation about churning (see comments in weaknesses part above)?
- Can the authors make a more compelling formulation of theorem 3.2 (see comments in weaknesses part)? It seems that neither this theorem nor the experiments actually show that the method prevents rank collapse.
- Isn't the method related to stochastic depth approaches in the sense that it allows the model to have variable depth with inputs? A data-dependent variable (but also recursive) depth approach was also proposed in [1]. Would it make sense similarly to use potential variable recurrent depth to vary the representation?

[1] Bae, S., Kim, Y., Bayat, R., Kim, S., Ha, J., Schuster, T., Fisch, A., Harutyunyan, H., Ji, Z., Courville, A. and Yun, S.Y., 2025. Mixture-of-recursions: Learning dynamic recursive depths for adaptive token-level computation. arXiv preprint arXiv:2507.10524.

**Limitations:**

None foreseen.

**Strengths And Weaknesses:**

**Strengths**
- The approach is rather simple and orthogonal to other training interventions that have been proposed to mitigate plasticity loss.
- The rationale for the intervention is argued through important metrics htat have been shown to correlate with plasticity loss.
- The proposed architectural change is in line with interventions like residual connections and gating mechanisms that had a large success is deep learning. It is a reasonable change in itself and mitigating plasticity loss is a strong argument for it.
- Experimental results show that the method performs clearly better than usual architectures on the proposed tasks. It may not particularly better than other training interventions but it is also independent of those and simpler to implement and study.
- The authors present a comprehensive set of experiments and metrics.

**Weaknesses**
- The claimed benefits (about churning and ntk rank collapse) are not clearly observed
  - In Fig. 5, it would be great to have an ablation with interplayers without dropout because it seems that dropout is essential to the success of the method (Fig 10)
  - In Fig 8 left bottom figure and in Fig 11 it seems that interplayers is not actually capable of prevent the rank collapse.
- The theory is a bit strangely formulated. For example the fact that $\text{rank}(N_{IL}) \geq \text{rank}(N_{interp})$ is true whether the variance of the interpolation weights is positive or not according to the proof (correct me if I'm wrong). I suppose what the authors want to point is that if the variance of the interpolation weights is positive, then $\text{rank}(N_{interp})$ is lower bounded so that you don't get collapse but then maybe a link between $\text{rank}(N_{interp})$ and the variance would be best. For now theorem 3.2 really does not show that rank collapse is avoided.

---

> ### Author Rebuttal · Authors · 2026-03-31
>
> The authors are thankful to the reviewer for your insightful and detailed comments.
>
> ---
> ### W1/Q1. (Churn Ablation)
>
> Thank you for your suggestion. We ran an additional experiment for InterpLayers without dropout and measured churn at the end of training by averaging the last five logged churn values across all conditions and seeds (10). We found that this case still exhibits substantially lower churn than a standard baseline without dropout (538.2 ± 132.2 vs. 7396.3 ± 997.7). Dropout reduces churn for both methods; InterpLayers with dropout obtain lower churn (29.5 ± 3.5) than the standard variant (285.7 ± 37.2). While dropout is helpful to lower churn, it is not the sole driver of this effect, since InterpLayers alone show substantial churn reduction compared to the standard variant.
>
> ---
> ### W2/Q2. (Rank Collapse)
>
> Our original analysis only examined the PPO heads to study the final effect of InterpLayers, whereas their immediate effect, in the encoder was left out. For a more proper examination, we ran an additional encoder-side NTK analysis using the same type of NTK construction adopted from Tang et al. (2025). For a given network, we collect gradient vectors across minibatches, stack them, and form the empirical kernel as a gradient-dot-product matrix. We then summarize its spectrum using the 99% mass approximate rank, i.e. the smallest $k$ such that the top-$k$ singular values explain 99% of the total spectral mass. Compared with our previous analysis, this methodology provides more precise information about rank preservation in the network.
>
> In this revised analysis, we found InterpLayers+Dropout achieves the strongest rank preservation. Since each minibatch has a size of 32, the maximum possible rank is 32. In the encoder, InterpLayers+Dropout reaches a 99%-mass approximate rank of 27.86 ± 0.37, compared with 25.77 ± 0.57 for Standard+Dropout, 18.33 ± 0.50 for SSP+LN, 16.51 ± 0.59 for Highway, 10.31 ± 0.98 for ResNet-like, and 14.25 ± 0.65 for the Standard baseline.
>
> We revised our manuscript accordingly by replacing the old NTK analysis on the PPO heads with the new encoder-side NTK analysis. These results support our theoretical prediction regarding the favorable properties of InterpLayers to prevent rank collapse.
>
> ---
> ### W3/Q2. (NTK Theorem)
>
> Thank you for the insightful comment! In Theorem 3.2, non-zero variance of interpolation weights $z(x)$ does not, on its own, guarantee a non-zero eigenvalue or show that rank collapse is avoided. We have revised the theorem to remove the variance-based sufficient condition and replaced it with an explicit Jacobian-feature formulation.
>
> For a finite sample set $\\{x_i\\}_{i=1}^n$, let
>
> $$
> J_p(x_i):=\\frac{\\partial h_{\\mathrm{proj}}(x_i)}{\\partial \\theta_p},\\qquad
> J_z(x_i):=\\frac{\\partial z(x_i)}{\\partial \\theta_z},
> $$
>
> and
>
> $$
> u_i:=\\operatorname{vec}\\!\\left(z(x_i)\\odot J_p(x_i)\\right),\\qquad
> v_i:=\\operatorname{vec}\\!\\left(J_z(x_i)\\odot\\big(h_{\\mathrm{proj}}(x_i)-h_{\\mathrm{ref}}(x_i)\\big)\\right).
> $$
>
> If $U$ and $V$ collect the row vectors $u_i^\\top$ and $v_i^\\top$, respectively, then the revised theorem shows
>
> $$
> N_{\\mathrm{IL}}=UU^\\top+VV^\\top,\\qquad N_{\\mathrm{interp}}=VV^\\top.
> $$
>
> Thus $N_{\\mathrm{interp}}$ is PSD, and
>
> $$
> \\operatorname{rank}(N_{\\mathrm{IL}})\\ge\\operatorname{rank}(N_{\\mathrm{interp}})=\\operatorname{rank}(V).
> $$
>
> and therefore contributes non-trivial rank whenever
>
> $$
> \\sum_{i=1}^n \\|v_i\\|_2^2>0,
> $$
>
> implying $\\operatorname{rank}(N_{\\mathrm{interp}})\\ge 1$. More generally, if $v_{i_1},\\dots,v_{i_r}$ are linearly independent, then $\\operatorname{rank}(N_{\\mathrm{interp}})\\ge r$.
>
> The revised theorem no longer rely on the variance of $z(x)$. Instead, it shows that the interpolation neurons contribute a non-trivial rank under explicit Jacobian-feature conditions. Note the revised manuscript now uses the variance of $z(x)$ only as an empirical proxy, not as the theorem's formal sufficient condition.
>
> ---
> ### Q3. (Stochastic Depth Approaches)
>
> This is an interesting point. InterpLayers are related to adaptive-computation approaches only at a high level, in the sense that both use the input to adapt how the model processes information. That said, InterpLayers are mechanistically different from Mixture-of-Recursions [1]. In MoR, different tokens are assigned different recursive depths, whereas in InterpLayers the network depth remains fixed and the representation is modulated through feature-wise interpolation between a fixed reference pathway and a learned projection pathway. Our method is therefore an interpolation-based architecture for selective adaptation, rather than a recursive or stochastic-depth approach. We will clarify this distinction and include [1] in the Related Works in our revision.
>
> ---
> We believe to have addressed all reviewer's concerns. If there are any further questions or concerns we will be happy to address them.

---

> > ### Author Rebuttal · Reviewer_5AjA · 2026-04-02
> >
> > W1: Thanks that's great.
> > W2: I think the real ntk should be computed by using the jacobian of the network (so removing the loss layer). What the authors have shown is more that the covariance of the gradients has its rank preserved (which is still a good argument for their method!).
> > W3: Excellent, it's much better and easy to read.
> >
> > The authors have completely solved my concerns. I'll increase my score to accept.

---

> > > ### Author Response · Authors · 2026-04-03
> > >
> > > Thank you for acknowledging our rebuttals! We are very glad that we have addressed all your concerns. Your comments helped us a lot to improve the theoretical foundation and empirical analysis of InterpLayers!

---

### Official Review · Reviewer_gfuE · 2026-03-11

**Soundness:** 3
**Presentation:** 3
**Significance:** 3
**Originality:** 3
**Overall Recommendation:** 4
**Confidence:** 3

**Summary:**

The paper proposes InterpLayers - an architectural mechanism for mitigating plasticity loss in continual reinforcement learning. The core idea is to replace standard layers with a dual-pathway module with a learnable projection pathway, using input-dependent interpolation weights. The authors discuss the area of continual reinforcement learning under non-stationary task sequences, where plasticity loss is associated with representational drift (churn), NTK rank collapse, and reduced adaptability over time. The core objective concerns whether a lightweight architectural modification can preserve plasticity without relying on external optimization tricks such as resets, shrink-perturb schedules, or explicit regularization.

The paper provides two main theoretical claims. First, it derives an upper bound on first-order output variability and uses this to argue that churn growth is controlled. Second, it derives an NTK decomposition and argues that the interpolation pathway provides a persistent non-zero rank contribution under a variance assumption on the interpolation weights. Empirically, the method is evaluated on four ProcGen environments under three distribution shifts: permute, window, and expand. The main finding is that InterpLayers outperform standard and gated architectural baselines and achieve performance comparable to the strong algorithmic baseline SSP+LN, while also being combinable with other methods.

**Compliance With Llm Reviewing Policy:**

Affirmed.

**Final Justification:**

The paper proposes InterpLayers for mitigating plasticity loss in continual reinforcement learning. The design is well-motivated. Experiments across multiple environments and shifts show competitive performance with strong baselines without requiring optimizer-level interventions. The rebuttal improves clarity by correcting theoretical claims, adding encoder-side NTK analysis, and refining the role of dropout and plasticity metrics.

However, limitations remain. Theoretical results provide only partial support and do not establish long-term stability. The strongest results depend on dropout, introducing an additional tuning factor. Empirical results largely show parity rather than clear improvement. The measurement of plasticity loss still relies on proxies and task performance rather than a well-defined metric. Therefore I maintain my score of *weak accept*.

**Key Questions For Authors:**

1. How is plasticity loss formally measured in the experiments? Is there a metric that directly quantifies loss of adaptability rather than relying only on task accuracy?

2. Theorem 3.2 appears to require more than non-zero variance in $z(x)$ to guarantee non-zero rank of $N_{interp}$. What additional conditions are actually sufficient?

3. The “bounded churn” corollary yields $C_T \leq B^2T^2$, which is a time-growing bound rather than a uniform bound. Can you revise the claim language to distinguish finite-horizon polynomial growth control from true asymptotic boundedness? If you believe a stronger statement holds, what assumptions are needed?

4. Appendix J.3 measures NTK only in the PPO heads, not the encoder where the interpolation mechanism acts most directly. Do you have any evidence (empirical or conceptual) that the same non-collapse behavior holds in the encoder gradients?

5. The main experimental variant uses dropout on the projection path, and Appendix J.2 shows sensitivity to dropout rate. How should readers interpret the claim that the method requires no additional hyperparameters, given that the best reported version depends on this choice?

**Limitations:**

No.

The paper does discuss extension to larger networks and notes that the NTK analysis is limited in scope, which is helpful. However, it should more clearly acknowledge three limitations: the restricted scope of the NTK measurement, the fact that the strongest empirical variant depends on dropout, and the gap between the actual theorem statements and the broader wording used in the main text of paper.

**Strengths And Weaknesses:**

**Strengths**

1. Plasticity loss in continual RL is an active topic and most current methods indeed act at the optimization level. A clean architectural intervention that is orthogonal to those methods is a meaningful contribution. The paper motivates this well and positions the method appropriately against reset-based and regularization-based approaches.

2. the proposed module is simple and reasonably elegant. The formulation is easy to understand and the reference/projection split gives a plausible mechanism for trading off stability and adaptation. The convolutional extension is also straightforward and implementable.

3. The evaluation spans four ProcGen tasks and three different distribution-shift regimes, with 10 seeds. it includes comparisons to standard networks, Highway networks, ResNet-like baselines, and SSP+LN. The main result is not that InterpLayers dominate SSP+LN, but that they are broadly competitive while requiring no optimizer-side intervention. That is a credible empirical claim.

**Weaknesses**

1. The proof in Appendix A.1 is correct, but it does not establish anything close to long-term stability by itself; it only gives a per-step first-order sensitivity bound. More importantly, the “bounded churn” corollary is overstated. The appendix derives
$C_T \leq B^2T^2$,
which is quadratic growth in training time, not a uniform bound independent of time. So the phrase "bounded churn" is true only in the weak sense that finite-time churn is upper-bounded by a function of $T$. It does not show that churn remains controlled asymptotically, and it certainly does not prove convergence or non-divergence as implied in the paper.

2.The NTK decomposition is plausible, and the rank lower bound $rank(N_{IL})\geq rank(N_{interp})$ is acceptable once both terms are PSD. But the key step claiming that non-zero variance in some coordinate of $z(x)$ guarantees a non-zero eigenvalue in $N_{interp}$ is not fully justified. The proof also states that if $z(x)$ collapses to a constant vector then interpolation gradients vanish "since $\nabla _{\theta _z} z(x) $ is zero almost everywhere after saturation", but constancy across samples does not, by itself, imply saturation or zero gradient with respect to gate parameters.

3. Related empirical weakness is that the NTK validation is much weaker than the main text suggests. Appendix J.3 explicitly says that, due to computational constraints, NTK is measured only in the PPO heads rather than throughout the entire network, and the reported result is that all variants maintain effective rank in late training. That is informative, but it is not a strong empirical validation of the architectural theorem, because the theorem is really about the interpolation mechanism in the representation stack, whereas the measurement is restricted to a small downstream part of the model.

4. The main model used in experiments is not bare InterpLayers; it is the conv-only InterpLayer encoder plus dropout on the projection pathway, with the dropout rate selected to be 0.05 after an appendix ablation. Appendix J.2 further shows that dropout has a real effect and involves a trade-off between plasticity retention and raw performance. So while InterpLayers themselves may be hyperparameter-light, the strongest experimental version still relies on a nontrivial design choice and tuning dimension. That does not invalidate the method, but the framing should be softened.

5. The conceptual distinction between plasticity loss and catastrophic forgetting is not always clearly formalized. The paper would benefit from a more precise operational definition and measurement protocol.


I believe that the paper gives meaningful contribution. The method is interesting, easy to implement, and likely useful to people working on plasticity loss in CRL. But the empirical outcome is mostly parity with SSP+LN rather than a clear advance over current strong methods. In Table 1, SSP+LN remains better in many settings, even though InterpLayers win some cases and are competitive overall.

---

> ### Author Rebuttal · Authors · 2026-03-31
>
> The authors are thankful to the reviewer for the insightful and detailed comments.
>
> ---
> ### W1/Q3. (Churn Proof)
>
> Thank you for your careful review of our proof. In line with your suggestion, we have revised the text to explicitly state that the proof established a per-step growth bound of churn when applying InterpLayers. We have also toned down the main text to make this distinction clearer.
>
> ---
> ### W2/Q2. (NTK Theorem/Proof)
>
> We thank the reviewer for pointing this out. We agree that Theorem 3.2 is too strong. In the revision, Theorem 3.2 is no longer stated in terms of non-zero variance of $z(x)$, but it shows that the interpolation contribution is a PSD Gram term whose rank is determined by the rank of the interpolation-feature matrix. Therefore, the variance of $z(x)$ is no longer used as a sufficient condition. We also removed the statement that a constant gate vector across samples implies saturation, since constancy alone does not imply vanishing gradients. See the response to W3/Q2 of 5AjA for the full revised theorem.
>
> ---
> ### W3/Q4. (NTK Measure)
>
> Thank you for highlighting the important point. We agree that, to properly assess the InterpLayer effects, the NTK should be measured in the encoder rather than in the PPO heads. Reflecting the reviewer's comments, we ran a new encoder-side analysis following Tang et al. (2025), summarized using the 99\% mass approximate rank. Critically, InterpLayers+Dropout achieves the strongest encoder-side rank preservation among all compared methods. We will revise the manuscript and replace the original PPO-head analysis with the encoder-side analysis in Appendix J.3. The full methodology and results are in the answer to W2/Q2 of 5AjA.
>
> We would like to share a conceptual insight in this regard. The gradient decomposition in Eq. 10 shows that the interpolation pathway contributes an additional encoder-side term through
>
> $$
> \nabla_{\theta_z} z(x) \odot \big(h_{\text{proj}}(x) - h_{\text{ref}}(x)\big)
> $$
>
> Our revised Theorem 3.2 uses this to state the rank preserving behavior. The new encoder-side NTK analysis is therefore aligned with both the architecture and the revised theoretical formulation. We will revise Appendix J.3 accordingly and replace the old PPO head-only analysis with the new encoder-side results.
>
> ---
> ### W4/Q5. (Dropout)
>
> In the seminal work by Goodfellow et al. [1], dropout was shown to help mitigate catastrophic forgetting. While these findings hint that dropout may reduce plasticity loss, the effect was not fully explored (examined only 2 tasks). We observe similar empirical evidence. For the standard baseline, incorporating dropout significantly improves performance. The results can be interpreted as corroborating the benefits of dropout in continual learning under current training and architectural paradigms.
>
> Furthermore, in Appendix H.1 of the manuscript, we provide a mathematical analysis of the effect of dropout on the projection pathway of InterpLayers, showing that it increases the activation variance of that pathway, which in turn influences the representational gap $D$, defined in Theorem 3.1. This way, dropout guarantees that $D$ retains some variance, stabilizing the gradients of the interpolation neurons.
>
> [1] Goodfellow, I. J., Mirza, M., Xiao, D., Courville, A., & Bengio, Y. (2013). An empirical investigation of catastrophic forgetting in gradient-based neural networks. arXiv preprint arXiv:1312.6211.
>
> ---
>
> ### W5. (Plasticity Loss Definition)
>
> We define *plasticity loss* as the gradual decline in a neural network’s ability to learn new tasks over time, i.e., a decrease in raw performance on new tasks despite comparable complexity and training information. This forms the main focus of our work. In contrast, *catastrophic forgetting* refers to the performance decrease on previously learned tasks when the model is updated with new tasks. We will clarify our definition of plasticity loss in the Introduction.
>
> ---
> ### Q1. (Measuring Plasticity Loss)
>
> In our experiments, we apply a performance-based metric to quantify plasticity loss, defined as the relative decrease in policy performance on new tasks compared to the performance achieved on the initial task. We will provide a formal mathematical definition in the Appendix.
>
> Since there is no standardized metric to quantify network adaptability, we follow the recent works of Lyle et al. (2024) and Tang et al. (2025) which showcase that NTK rank and churn correlate with plasticity loss, and can serve as proxies.
>
> ---
> We believe to have addressed all reviewer's concerns. If there are any further questions or concerns we will be happy to address them.

---

> > ### Author Rebuttal · Reviewer_gfuE · 2026-04-02
> >
> > Thank you for the detailed and thoughtful rebuttal. The revisions and clarifications address several of the concerns raised in the original review. However there are some key concerns that remain.
> > 1. The discussion around dropout clarifies its role as a complementary mechanism rather than a required component of the architecture. However, the concern remains that the strongest empirical results depend on this design choice, which introduces an additional tuning dimension. While this does not undermine the contribution, the framing that the method is entirely hyperparameter-free should be interpreted with some caution.
> >
> > 2. The rebuttal does not fully address the need for a more precise operational definition of plasticity loss and its measurement. The current evaluation still relies primarily on task performance under distribution shift. While this is standard, a more direct metric of adaptability  would strengthen the claims.
> >
> > 3. lastly, i feel that empirical results are showing parity rather than clear improvement over strong baselines.
> >
> > Overall, the paper presents a well-motivated and technically sound architectural contribution. However, I feel there are certain areas where the paper could improve. And therefore, I wish to maintain my original evaluation.

---

> > > ### Author Response · Authors · 2026-04-06
> > >
> > > Thank you for acknowledging our rebuttal and our architectural contribution! We are glad for the opportunity to follow up on the reviewer’s concerns.
> > >
> > > 1. (Dropout)
> > >
> > > As the reviewer points out, our strongest empirical variant depends on the dropout choice, so we will soften the ‘hyperparameter-free’ framing. We view dropout as a structured regularizer of the projection/interpolation dynamics in InterpLayers. As discussed in Appendix H.1, dropout is applied only to the projection pathway, where it increases the activation variance and affects the representational gap $D=\\lVert h_{\\mathrm{proj}}(x)-h_{\\mathrm{ref}}(x) \\rVert_2$ in Theorem 3.1. Here, dropout prevents the projection from aligning too closely with the reference pathway, which sustains a non-degenerate $D$. This ensures that the interpolation pathway stays relevant to the output and its gradients remain active. In other words, dropout  supports how InterpLayers balance stability and adaptation, rather than acting as a generic external regularizer. This interpretation is consistent with our findings in Fig. 10: across non-zero dropout rates, performance retention is relatively robust, whereas removing dropout lowers it. However, as dropout decreases effective network capacity, higher rates lead to lower raw performance. A dropout rate of 0.05 provides the best balance of retention and raw performance. We observe this effect in practice, where the introduction of dropout to the projection pathway of InterpLayers has a small cost in terms of raw performance but reduces churn and gradient magnitude while increasing empirical NTK rank. We will revise the manuscript to make it clearer that the strongest variant depends on a tunable dropout choice and that dropout interacts with InterpLayers through the dynamics described in Appendix H.
> > >
> > > 2. (Plasticity Metrics)
> > >
> > > Thank you for your insightful comment. We agree that a metric for adaptability independent of task accuracy would strengthen the measure of plasticity loss in continual learning. Followingly, we performed an additional literature review on current metrics of adaptability applied in continual learning. Earlier works of Lyle et al. [1] and Elsayed & Mahmood [2] have introduced plasticity metrics that do not rely on task accuracy. Lyle et al. proposed a probe-based plasticity metric that directly measures how well a network can adapt to a new probe task after a fixed number of optimization steps. Their work also suggests that known proxies of plasticity loss, like weight norm, weight rank, and number of dead neurons, are not good empirical metrics of network adaptability. Elsayed & Mahmood introduced ‘sample plasticity’, which quantifies the agent's immediate ability to improve on the current sample after one update. Sample plasticity is defined as the normalized reduction in loss from before to after the update. This quantity is averaged over an evaluation window to track the loss/maintenance of plasticity.
> > >
> > > We implemented both of these metrics and re-ran our CoinRun experiments. In our setting, sample plasticity shows a weak but consistent positive correlation with both raw performance and its retention. The probe-based plasticity shows much lower correlation with either value. These preliminary results suggest that sample plasticity may be a useful candidate for measuring plasticity loss. We will include this analysis in our appendix and discuss these metrics in our discussion, pointing to sample plasticity as a possible anchor point for future work investigating plasticity metrics.
> > >
> > > We also performed an additional correlation analysis using the empirical proxies we employ in the paper, churn and NTK rank. Both of these show a strong correlation with raw performance and retention. In particular, churn has the strongest correlation with retention (Pearson correlation of -0.643) and raw performance (-0.634) among all metrics. From these results, our analysis implies that churn is an effective non-task accuracy metric to measure plasticity.
> > >
> > > We will revise the manuscript, adding a discussion regarding network adaptability metrics, highlighting how their combination with accuracy metrics is important while studying continual learning, and making the relationship between churn and plasticity loss more explicit.
> > >
> > > [1] Lyle, Clare, et al. "Understanding plasticity in neural networks." ICML. PMLR, 2023.
> > >
> > > [2] Elsayed, Mohamed, and A. Rupam Mahmood. "Addressing loss of plasticity and catastrophic forgetting in continual learning." arXiv:2404.00781 (2024)
> > >
> > > 3. (Experimental Results)
> > >
> > > We acknowledge that InterpLayers show comparative results against SOTA baselines like SSP+LN. SSP+LN shrinks and perturbs the weights at every optimizer step. On the other hand, InterpLayers do not require this intervention, providing an alternative solution on the architectural level. Our main goal and contribution is to show that architectural solutions to plasticity loss can compete with current SOTA algorithmic solutions.

---

### Official Review · Reviewer_qvQP · 2026-03-18

**Soundness:** 3
**Presentation:** 3
**Significance:** 2
**Originality:** 2
**Overall Recommendation:** 4
**Confidence:** 4

**Summary:**

This paper addresses plasticity loss in continual reinforcement learning. The authors introduce InterLayers, an architectural modification that introduces a learnable interpolator (e.g. convex combination) between essentially a skip connection and the standard learnable pathway of the network. This approach is orthogonal and distinct from existing work that is primarily algorithmic rather than architectural, such as, resets, regularization schemes, modifications to the optimization algorithm, normalization and clipping. The authors argue that these methods can require additional information about the non-stationarity, hyper parameters to tune, and are implemented as external modifications to the optimization framework, while InterLayer is free from such hyper parameters and tuning requirements. Demonstrate the following theoretical properties: polynomially bounded representational drift and preservation of gradient diversity. Empirical results corroborate the theoretical claims and the proposed method shows competitive performance against the evaluated competitor architectures and methods.

**Compliance With Llm Reviewing Policy:**

Affirmed.

**Final Justification:**

The author's rebuttal has largely resolved my concerns. I think that slightly more exhaustive experiments may be necessary, but I am happy to raise my score given the novelty of introducing an architectural intervention in juxtaposition to existing neuron reset or regularization-based intervention for plasticity loss.

**Key Questions For Authors:**

- How sensitive is InterpLayers with respect to the learning rate and optimizer parameters? While this method is proposed to be hyper parameter free and require no tuning, this may not be entirely true if additional learning rate tuning is required, for instance.

**Limitations:**

Authors do not state limitations. Yes, there is an impact statement.

**Strengths And Weaknesses:**

Strengths:
- The introduction of InterpLayers in Section 3.2, equations 1-4, is very clear.
- The theory is clear to follow and the Theorems 3.1 and 3.2 provide sufficient theoretical support for InterpLayers.
- Experiments show positive efficacy of the proposed method.

Weaknesses:
- The experiments in section 4.2, namely Figures 2 and 3 and Table 1 show that InterpLayers is essentially tied as second best with Standard + SSP + LN. These are rather mixed results and given the limited number of competitor architectures/methods, it is hard to discern to efficacy of InterpLayers.
- For the ablation experiment shown in Figure 4, it would be useful to include the other baselines with these additional algorithmic interventions for fairness of comparison. This ablation only demonstrates that additional algorithmic interventions improve performance, which is unsurprisingly in continual learning.
- While Figure 5 shows reduced Churn for InterpLayers, we see that competitor methods also achieve reduced churn. Coupled with earlier results, these results do not show that InterpLayers is superior to existing methods for continual learning.
- It would be useful to evaluate additional competitor methods. For instance, how do reset based methods like CBP, SNR, ReDO, ore regularization techniques like L2 Init, spectral regularization, etc compare? While these methods algorithmic in comparison to InterpLayers, a point is made in this paper that InterpLayer is not an algorithmic intervention and thus comes with several benefits such as no parameter tuning. Therefore, it would be useful to benchmark against other algorithmic interventions.
- Overall, the experiments appear rather limited and show mixed performance against the limited set of competitor architectures/methods.

Suggestions/Comments:
- In the Related Works on subsection, in addition to ReDO and Continual Backdrop, there is a third continual reset method called Self-Normalized Resets (SNR). SNR has a self-normalizing mechanism that limits the need for parameter tuning, which is partially counter to the comment on line 055 and 076 on the second column. It may be useful to highlight this.
- I would suggest wording the introduction in a similar manner to my summary. Before reading equations 1-4 on page 3, it is not clear what exactly InterpLayer is and how it differs from a residual connection. Specifically, it would be useful to introduce InterpLayer as learning a convex interpolator between a skip connection and the learnable pathway of a neural network. The current one-sentence description is somewhat cumbersome.

---

> ### Author Rebuttal · Authors · 2026-03-31
>
> The authors are thankful to the reviewer for your insightful and detailed comments.
>
> ---
> ### W1/W5. (Experimental Results)
>
> We acknowledge that InterpLayers show comparative results with Standard+SSP+LN. While SSP+LN performs weight shrinking and perturbation at every optimizer step, InterpLayers reach comparable performance without such intervention. Our main contribution is thus to demonstrate that InterpLayer offer an alternative solution to plasticity loss next to SOTA algorithmic interventions. Notably, these results are achieved with lower inference cost: as shown in Table 2, our conv-only InterpLayers variant uses 49.7M forward FLOPs per inference step, compared to 67.5M for Standard+SSP+LN.
>
> To clarify and strengthen the robustness of our results, we ran additional experiments during the rebuttal on continual Gym Control and expanded the baseline set with ReDo, L2-Init, and a HyperNetwork-style architecture. The overall trends remain consistent! ReDo is a strong baseline but has lower retention than InterpLayers, L2-Init is weaker overall, and HyperNetwork-style networks perform well in mild shifts but show instability in stronger shifts (permute). InterpLayers remain competitive with the strongest algorithmic baselines (SSP+LN and ReDO) and consistently outperform the architectural ones.
>
> ---
> ### W2. (Additional Ablations)
>
> Thank you for this suggestion. To improve fairness in the comparison, we extend the ablation and apply LN and SSP+LN to the other architectural baselines as well, including a new HyperNetwork-style baseline.
>
> Overall, the same conclusion as for InterpLayers holds. Adding LN or SSP+LN to these architectures show mixed results. For Highway and ResNet-like, adding LN or SSP+LN leads to modest changes and offers no clear improvement in raw performance or retention. For the HyperNetwork-style baseline, these improve stability in some settings but do not resolve its low performance under the strongest shift (permute).
>
> We will revise the manuscript and include these results in the ablation study.
>
> ---
> ### W3. (Churn Analysis)
>
> Thank you for the insightful comment. We built on the hypothesis by Tang et al. (2025) that high churn correlates with plasticity loss metrics. First, we use this hypothesis to theoretically analyze InterpLayers through the lens of bounded representational drift (Theorem 3.1). Second, we observe that, across our experiments, the best-performing baselines also exhibit low churn. These findings indicate that reduced churn is important for mitigating plasticity loss. Since no standardized network metric currently exists for plasticity loss, we adopted churn as an empirical proxy. Meanwhile, as the reviewer rightly notes, this serves as a surrogate measure and does not necessarily yield stronger performance retention.
>
> ---
> ### W4. (Additional Baselines)
>
> Based on the reviewer’s suggestion, we compared InterpLayers against ReDo and L2-Init on CoinRun. Aggregated over expand, permute, and window, ReDo is slightly higher in raw performance (8.919 ± 0.042) than InterpLayers (8.826 ± 0.057), but InterpLayers show higher retention (1.025 ± 0.009 vs. 0.984 ± 0.006). L2-Init is considerably worse on both metrics (5.821 ± 0.066, retention 0.892 ± 0.017). Therefore, reset-based methods like ReDo are also strong baselines, while L2-Init is less competitive. Comparatively, the main baseline used in our work for comparison, SSP+LN, performs better than ReDo in both raw rewards performance (9.213 ± 0.030 vs. 8.919 ± 0.042) and retention (1.034 ± 0.006 vs. 0.984 ± 0.006).
>
> ---
> ### C1. (Additional Related Work)
>
> Thanks! We will discuss SNRs in our Related Works.
>
> ---
> ### C2. (Introduction Clarity)
>
> Adopting the reviewer's suggestion, we will revise the 3rd paragraph of the Introduction to present InterpLayers as learning a convex interpolator between a skip connection and the learnable pathway.
>
> ---
> ### Q1. (Sensitivity to Hyperparameters)
>
> We ran a learning-rate ablation for InterpLayers on CoinRun with `lr ∈ {5e-5, 1e-4, 5e-4}`. Our findings show that the performance on plasticity preservation is robust for these learning rate values while the raw rewards performance show higher variation. The average plasticity retention across the three shift types is 1.032 for 5e-5, 1.066 for 1e-4, and 0.991 for 5e-4. Thus, 1e-4 gives the best retention, but all variants show relatively robust results maintaining plasticity. For the raw reward performance, the average pre-shift return across conditions is 8.54 for 1e-4, compared with 8.03 for 5e-5 and 7.93 for 5e-4.
>
> Overall these results suggest that InterpLayers are reasonably robust to the learning rate in our setup, especially with respect to plasticity preservation. While 1e-4 performs best overall, there is no strong degradation across this non-trivial range of values.
>
> ---
> We believe we have addressed all reviewer's concerns. For any follow-up questions or clarifications, we would be happy to provide additional details.

---

> > ### Author Rebuttal · Reviewer_qvQP · 2026-04-05
> >
> > The author's rebuttal has largely resolved my concerns. I think that slightly more exhaustive experiments may be necessary, but I happy to raise my score given the novelty of architectural intervention rather than a neuron reset or regularization-based intervention for plasticity loss.

---

> > > ### Author Response · Authors · 2026-04-06
> > >
> > > Thank you for acknowledging our rebuttal! Your comments have improved the empirical scope of our work a lot! We are happy to follow up with additional experiments that we ran in order to further strengthen the robustness and performance of InterpLayers. All experiments used 10 seeds for each variant/condition.
> > >
> > > First, we ran an additional ablation for different optimizers to extend the evaluation of the robustness of InterpLayers. We used four different optimizers: Adam, AdamW, RMSProp, and SGD. For the first three optimizers, InterpLayers show stable raw performance and retention with the Adam variants having slightly better results than RMSProp (8.832 ± 0.052 / 1.025 ± 0.009 and 8.815 ± 0.053 / 1.031 ± 0.009 vs. 8.314. ± 0.066 / 1.039 ± 0.007). For SGD, we observed a strong performance decrease (6.000 ± 0.050 / 0.969 ± 0.012). However, we observed an even further performance decrease if we use Standard+Dropout with SGD (5.994 ± 0.050 / 0.945 ± 0.012), suggesting that this degradation is not specific to our method but rather a general limitation of the optimizer. These results indicate that InterpLayers are not sensitive to a specific optimizer choice, particularly in terms of performance retention, as the scores remain consistent across a range of commonly used optimizers.
> > >
> > > Secondly, we expanded the environments in which we evaluate the new baselines. We further evaluated ReDo, L2-Init, and a HyperNetwork-like model in the three ProcGen environments that we had used in the paper, namely FruitBot, Heist, and Jumper. Across these additional tasks, InterpLayers+Dropout remains strong overall and outperforms the new baselines on more challenging tasks like FruitBot and Heist. In Heist, InterpLayers+Dropout reach a raw performance of 6.102 ± 0.484, compared with 4.144 ± 0.496 for ReDo, 3.919 ± 0.478 for HyperNetwork-like, and 2.543 ± 0.429 for L2-Init. In FruitBot, we observe 4.515 ± 0.529, 4.096 ± 0.479, 3.388 ± 0.404, and -2.422 ± 0.040 for InterpLayers, ReDo, HyperNetwork-like, and L2-Init, respectively. In Jumper, all stronger methods perform in a similar range with InterpLayers+Dropout at 8.121 ± 0.041, ReDo at 8.358 ± 0.029, HyperNetwork-like at 8.292 ± 0.055, while L2-init is weaker at 3.952 ± 0.141. Overall, the additional results strengthen our core claim that InterpLayers provide a strong architectural approach for mitigating plasticity loss in continual learning.
> > >
> > > We will revise the manuscript to include these additional baselines in our empirical evaluation to further strengthen our manuscript.

---

### Decision · Program_Chairs · 2026-04-30

**Decision:**

Accept (regular)

**Comment:**

This paper studies plasticity loss in continual reinforcement learning and proposes InterpLayers, a simple architectural intervention that interpolates between a fixed reference pathway and a learnable projection pathway, aiming to preserve adaptability without relying on optimizer-level tricks such as resets, perturbation schedules, or explicit regularization. The reviewers generally found the idea meaningful and well motivated, especially because it offers an architectural alternative that is complementary to existing algorithmic methods. The overall review profile is supportive, roughly 5/4/4/4, with reviewers acknowledging that the method is simple, implementable, and competitive with strong baselines even when not clearly dominating them. The main concerns were that the empirical gains were often closer to parity than clear superiority over strong algorithmic baselines such as SSP+LN and ReDo, that some theoretical claims were initially overstated or imprecisely formulated, that the strongest empirical variant depended on dropout and thus was not entirely hyperparameter-free, and that the evaluation scope and architectural comparisons could be broadened. In the rebuttal, the authors addressed these concerns constructively by softening the theoretical claims, revising the NTK theorem, replacing the PPO-head NTK analysis with encoder-side measurements, adding churn ablations without dropout, clarifying the role of dropout and hyperparameter sensitivity, expanding comparisons to ReDo, L2-Init, and HyperNetwork-style baselines across additional environments, and providing a clearer operational distinction between plasticity loss and catastrophic forgetting. These additions were viewed positively, and multiple reviewers indicated that their main concerns were resolved and raised or maintained supportive scores. Overall, I find that the paper makes a meaningful and timely contribution by showing that a lightweight architectural modification can be competitive with state-of-the-art algorithmic interventions for mitigating plasticity loss, and I recommend Accept.